# Atmospheric Inverse Modeling via Sparse Reconstruction

Nils Hase[1], Scot M. Miller[2], Peter Maaß[1], Justus Notholt[3], Mathias Palm[3], and Thorsten Warneke[3]

[1]Center for Industrial Mathematics, University of Bremen, Bremen, Germany
[2]Department of Global Ecology, Carnegie Institution for Science, Stanford, CA, USA
[3]Institute of Environmental Physics, University of Bremen, Bremen, Germany

*Correspondence to:* N. Hase (nilshase@math.uni-bremen.de)

**Abstract.** Many applications in atmospheric science involve ill-posed inverse problems. A crucial component of many inverse problems is the proper formulation of a priori knowledge about the unknown parameters. In most cases, this knowledge is expressed as a Gaussian prior. This formulation often performs well at capturing smoothed, large-scale processes but is often ill-equipped to capture localized structures like large point sources or localized hot spots.

Over the last decade, scientists from a diverse array of applied mathematics and engineering fields have developed sparse reconstruction techniques to identify localized structures. In this study, we present a new regularization approach for ill-posed inverse problems in atmospheric science. It is based on Tikhonov regularization with sparsity constraint and allows bounds on the parameters. We enforce sparsity using a dictionary representation system. We analyze its performance in an atmospheric inverse modeling scenario by estimating anthropogenic US methane ($CH_4$) emissions from simulated atmospheric measurements.

Different measures indicate that our sparse reconstruction approach is better able to capture large point sources or localized hot spots than other methods commonly used in atmospheric inversions. It captures the overall signal equally well but adds details on the grid scale. This feature can be of value for any inverse problem with point or spatially discrete sources. We show an example for source estimation of synthetic methane emissions from the Barnett shale formation.

## 1 Introduction

Inverse problems are widespread in atmospheric sciences. The estimation of greenhouse gas sources and sinks is a prime example. Numerous studies combine observations of greenhouse gas concentrations in the atmosphere and inverse modeling to infer sources and sinks at the Earth's surface. Existing studies apply these techniques at municipal (e.g. Saide et al., 2011), regional (e.g. Zhao et al., 2009), continental (e.g. Miller et al., 2013) and global scale (e.g. Stohl et al., 2009). Inverse modeling estimates of greenhouse gas emissions are not only of scientific interest (e.g., to assess biospheric fluxes or improve process-based models). These estimates are also key for monitoring and evaluating greenhouse gas emissions regulations (U.S. National Research Council, 2010).

In almost all cases, these parameter estimation problems are ill-posed. Ill-posed means that small noise on the measurements can be amplified by the inversion leading to unrealistic estimates. Thus, special techniques are required for a stable inversion.

A Bayesian inversion is a common tool in atmospheric sciences that can handle the ill-posed nature of these problems (e.g. Rodgers, 2000). In Bayesian inversion, the unknown parameters are assumed to follow an a priori distribution. The observations are used to calculate an a posteriori distribution, which contains balanced information from the observations and the prior. The maximum of the a posteriori distribution is often used as a best estimate.

A classical approach is the use of a Gaussian prior, which often allows rapid calculations via analytical expressions. However, the Gaussian prior is known to return a best estimate that is a smoothed version of the true solution (Rodgers, 2000, ch. 3). It is well-suited to detect the overall process, but local structures such as large point sources are often smoothed out for ill-posed problems.

Other research areas solve inverse problems using Tikhonov regularization. Tikhonov regularization is formulated as an optimization problem. The functional to be minimized consists of a data fitting term and a penalty term that prevents overfitting. The classical choice of these terms is analogous to a Bayesian inversion with a Gaussian prior.

Recently, Tikhonov regularization with sparsity constraint has become a popular alternative to these classical inverse methods within a number of engineering fields. Several recent studies apply the approach to a variety of applications, including medical imaging, signal analysis and compressed sensing (see e.g. Hämäläinen et al., 2013; Knopp and Weber, 2013; Candès et al., 2011). All of these applications make use of the fact that the underlying process can be described as a localized signal in a suitable representation system. While the classical approach tends to smooth the true process (in any representation system), sparse reconstruction is designed to find such localized structures. Jin and Maass (2012) give a detailed summary of the mathematical advances with the sparsity constraint.

Only a handful of studies apply these modern inversion techniques to atmospheric sciences. Martinez-Camara et al. (2013) used a sparse reconstruction approach to estimate emissions of radioactive substances for the Fukushima accident, and Ray et al. (2015) analyzed fossil fuel carbon dioxide emissions in an idealized, synthetic data setup.

The goal of this paper is to show how sparse reconstruction techniques can improve flux estimates in an atmospheric inverse modeling scenario. We use a synthetic case study from Miller et al. (2014), a study that explores different inverse modeling methods that enforce non-negative surface fluxes. The setup considers anthropogenic methane emissions in the United States. We couple a sparse reconstruction approach with a positivity constraint.

The present study is organized as follows: First, we briefly introduce the atmospheric inverse modeling problem (Sect. 2). Section 3 gives an overview of inverse problems and introduces the concept of sparse reconstruction. We use a redundant dictionary representation system to sparsify the flux signal. The setup of the synthetic case study and the sparse dictionary reconstruction method are presented in Sect. 4. Estimates, error analysis and a comparison with state-of-the-art methods are shown in the results Sect. 5. We also analyze the sensitivity to emissions from an oil and gas drilling region before drawing conclusions.

Additional graphics, source code and a pseudocode of the sparse dictionary reconstruction method are included in the supplementary information.

## 2 Surface flux estimation using atmospheric inverse modeling

Existing studies employ a number of different techniques to quantify greenhouse gas surface fluxes (e.g. Hensen et al., 2013). Atmospheric inverse modeling (AIM) is an approach that relies on the knowledge of a proper atmospheric transport model to link surface sources and sinks to enhancements in atmospheric greenhouse gas concentrations. The idea is to invert the transport model and thus map atmospheric measurements to surface fluxes.

We use the WRF-STILT (Weather Research and Forecasting - Stochastic Time-Inverted Lagrangian Transport) model (Nehrkorn et al., 2010) to simulate atmospheric transport in this study, the same simulations used in Miller et al. (2013, 2014). WRF is a meteorology model (e.g. Skamarock et al., 2005), and STILT is a back-trajectory model (e.g. Lin et al., 2003; Gerbig et al., 2003). STILT releases an ensemble of imaginary particles at the time and location of an atmospheric measurement. The particles then travel backward in time and indicate where air masses were located before reaching the measurement location. STILT then uses the distribution of these particles to compute an upwind surface influence on the measurement, called footprint. The footprint quantitatively relates the surface fluxes to the atmospheric measurement (in units of atmospheric mixing ratio per unit of surface flux).

For a given emission field, $x$, the enhancement of the measurement above a known background level, $y_k$, can be simulated by integrating the product of footprint, $A_k$, and emissions over the Earth's surface area of interest, $\Omega$:

$$\boldsymbol{y} = \mathbf{A}\boldsymbol{x}, \quad \text{where} \quad [Ax]_k := \int_{\Omega} A_k(s)x(s)ds \tag{1}$$

and $s \in \Omega$ is the integration variable of location. The central question in AIM is how to determine a realistic flux field $x$ given (noisy) atmospheric measurements $\boldsymbol{y_\delta}$ and footprints $\mathbf{A}$, which means solving the inverse problem of Eq. (1). Apart from the inverse problem itself, AIM may involve a number of additional challenges, including but not limited to estimation of background concentrations and proper modeling of atmospheric transport and chemistry. The present article only adresses the solution of the inverse problem.

## 3 Mathematical background of inverse problems

In this section we provide some mathematical background for inverse problems and how the approach developed in this article is related to commonly used inverse methods. We formulate the AIM problem as a parameter optimization problem, which is based on norm notation. Thus, we define

$$\|\boldsymbol{z}\|_2 := \sqrt{\sum_k |z_k|^2} \quad \text{and} \quad \|\boldsymbol{z}\|_1 := \sum_k |z_k|.$$

Both norms measure the length of a vector $\boldsymbol{z}$, where $\boldsymbol{z}$ is a vector of any quantity. The 2-norm is the standard norm in most fields of study. The 1-norm is a central concept in sparse reconstruction, as discussed in Sect. 3.3.

## 3.1 Ill-posed inverse problems

Inverse problems arise when the quantity of interest cannot be measured directly. Instead, another quantity $\boldsymbol{y}$ is measured that is related to the unknown parameters $\boldsymbol{x}$ by a forward model $\mathbf{F}$. The forward model maps from parameter space $X$ to measurement space $Y$. For most problems, $X = \mathbb{R}^n$ and $Y = \mathbb{R}^m$. The forward problem is to calculate simulated measurement data $\boldsymbol{y}$ from known parameters $\boldsymbol{x}$ by evaluating the potentially nonlinear forward model, $\boldsymbol{y} = \mathbf{F}(\boldsymbol{x})$. Estimating realistic parameters that explain the given measurements means solving the inverse problem.

The problem of finding parameters that best explain noisy measurements $\boldsymbol{y_\delta}$ in a least squares sense is equivalent to solving

$$\boldsymbol{x}^* = \arg\min_{\boldsymbol{x} \in X} \frac{1}{2} \|\mathbf{F}(\boldsymbol{x}) - \boldsymbol{y_\delta}\|_2^2. \tag{2}$$

Often the inverse problem is ill-posed. This means that the minimizer $\boldsymbol{x}^*$, if one exists, might be nonunique and particularly that the inversion is unstable. Unstable denotes that small changes in the measurements result in large changes in estimated parameters. In the real world, measurements are never exact. A common assumption is that the noisy measurements $\boldsymbol{y_\delta}$ can be split up into exact data, $\boldsymbol{y} \in Y$, and noise, $\boldsymbol{\delta} \in Y$, such that $\boldsymbol{y_\delta} = \boldsymbol{y} + \boldsymbol{\delta}$. The exact data is defined by the true parameters $\boldsymbol{x}^+$ via the underlying forward model, i.e. $\boldsymbol{y} = \mathbf{F}(\boldsymbol{x}^+)$. In this definition, the noise includes errors from the measurements, the forward model and numerical approximations. Solving Eq. (2) means fitting the parameters to the exact data and the noise. In ill-posed problems, the retrieved parameters are very sensitive to data, so the true solution $\boldsymbol{x}^+$ is typically far away from the least squares solution $\boldsymbol{x}^*$. Thus, the inverse mapping using Eq. (2) is not suitable for ill-posed problems (see the supplementary information).

## 3.2 Tikhonov regularization

The inversion using Eq. (2) is unstable for ill-posed problems. Tikhonov regularization, by contrast, stabilizes the inversion by adding a convex penalty function $\phi : X \to \mathbb{R}$ (see e.g. Hansen, 2010; Louis, 1989)

$$\boldsymbol{x}^* = \arg\min_{\boldsymbol{x} \in X} \frac{1}{2} \|\mathbf{F}(\boldsymbol{x}) - \boldsymbol{y_\delta}\|_2^2 + \alpha\phi(\boldsymbol{x}). \tag{3}$$

Classical Tikhonov regularization uses $\phi(\boldsymbol{x}) = \frac{1}{2}\|\boldsymbol{x} - \boldsymbol{x_a}\|_2^2$ with $\boldsymbol{x_a} = \boldsymbol{0}$. The regularization parameter $\alpha$, with $\alpha > 0$, weights the data fitting and the penalty term. A greater value forces the solution to stay close to the a priori solution $\boldsymbol{x_a}$, while a small value results in a better model-data fit. A number of methods are available to automatically choose a balancing regularization parameter (see e.g. Reichel and Rodriguez, 2012). We use Morozov's discrepancy principle (see Eq. (15) in Sect. 4.2), which requires knowledge of the noise level $\|\boldsymbol{\delta}\|$. A suitable regularization parameter prevents overfitting of the estimated parameters $\boldsymbol{x}^*$ to the noisy data via the forward model.

If more detailed noise characteristics are known, these can be introduced by adaptation of the data fitting term. In case of Gaussian noise, penalized, weighted least squares

$$\boldsymbol{x}^* = \arg\min_{\boldsymbol{x} \in X} \frac{1}{2} \|\mathbf{L}_\delta(\mathbf{F}(\boldsymbol{x}) - \boldsymbol{y_\delta})\|_2^2 + \alpha\phi(\boldsymbol{x}) \tag{4}$$

with a noise covariance $\mathbf{R}$ and $\mathbf{R}^{-1} = \mathbf{L}_\delta^t \mathbf{L}_\delta$ best incorporates this information. The covariance is especially useful to weight measurements with different uncertainties. For a suitable penalty function, this approach translates to Bayesian inverse modeling (see Sect. 3.6).

### 3.3 Choice of the penalty function

We solve the inverse problem using Tikhonov regularization, Eq. (4). Information about the measurement noise is introduced in the data fitting term of the Tikhonov functional, and prior information about the unknown parameters is formulated in the penalty function $\phi$. In the absence of any prior information, the classical Tikhonov approach uses a 2-norm penalty with zero a priori, $\phi(\boldsymbol{x}) = \frac{1}{2}\|\boldsymbol{x}\|_2^2$. Among all possible solutions, it chooses the solution that is closest to the origin, i.e. $(0,...,0) \in X$ but still reproduces the data. Proximity to the origin means that the solution is simple. In particular, it prevents the oscillating

behaviour of the parameters as encountered when the inversion is unstable (e.g. Hansen, 2010, ch. 4). If an a priori estimate $x_a$ of the parameters is available, it can be included by setting $\phi(\boldsymbol{x}) = \frac{1}{2}\|\boldsymbol{x} - \boldsymbol{x_a}\|_2^2$.

Sometimes it can be useful to penalize the components of the parameter vector $\boldsymbol{x}$ differently. This results in a weighted 2-norm penalty: $\phi(\boldsymbol{x}) = \frac{1}{2}\|\mathbf{L_a}\boldsymbol{x}\|_2^2$. In a Bayesian inversion setup $\mathbf{Q} = \frac{1}{\alpha}(\mathbf{L_a^t L_a})^{-1}$ gives the covariance of a multivariate Gaussian a priori distribution. Diagonal elements in $\mathbf{L_a}$ weight the parameters while off-diagonal entries correlate parameters

resulting in smoother estimates in case of positive correlation and vice versa.

A large number of methods are available to solve optimization problems of the type (see e.g. Rodgers, 2000, ch. 5)

$$\boldsymbol{x}^* = \arg\min_{\boldsymbol{x} \in X} \frac{1}{2}\|\mathbf{L}_\delta(\mathbf{F}(\boldsymbol{x}) - \boldsymbol{y_\delta})\|_2^2 + \frac{\alpha}{2}\|\mathbf{L_a}(\boldsymbol{x} - \boldsymbol{x_a})\|_2^2. \tag{5}$$

The minimizer $\boldsymbol{x}^*$, if it exists (see Sect. 3.4), can also be interpreted as a maximum a posteriori solution to a Gaussian prior with Gaussian noise in a Bayesian inversion framework (see Rodgers, 2000, ch. 3).

Localized structures like point sources or edges in the true solution $\boldsymbol{x}^+$ are smoothed out by regularization with the 2-norm and thus disappear in the estimate $\boldsymbol{x}^*$.

The sparsity contraint has become very popular for regularization of inverse problems over the last decade. The 1-norm is used to constrain parameters instead of taking the 2-norm as a penalty function. This results in the optimization problem

$$\boldsymbol{x}^* = \arg\min_{\boldsymbol{x} \in X} \frac{1}{2}\|\mathbf{L}_\delta(\mathbf{F}(\boldsymbol{x}) - \boldsymbol{y_\delta})\|_2^2 + \alpha\|\boldsymbol{x}\|_1. \tag{6}$$

The constraint produces solutions with only a few nonzero components, which are called sparse solutions. Replacing the 2-norm by the 1-norm adds a greater penalty on small components in the solution and favors the inclusion of larger components within the solution. The effect is that the 1-norm constraint selects a sufficient number of components to explain the data while setting the others to zero. In constrast, the 2-norm penalty uses all the components to reproduce the data and avoids large components. Figure 1 illustrates why parameters that are not sufficiently constrained by the data are set to zero when using the

sparsity constraint. Due to this property, the method in Eq. (6) is called sparse reconstruction.

Flux fields with sporadic local hot spots are a prime example of sparse signals and sparse reconstruction is well-suited to identify such sparse but nonsmooth signals. However, if the true solution $\boldsymbol{x}^+$ is not sparse, reconstruction by Eq. (6) will

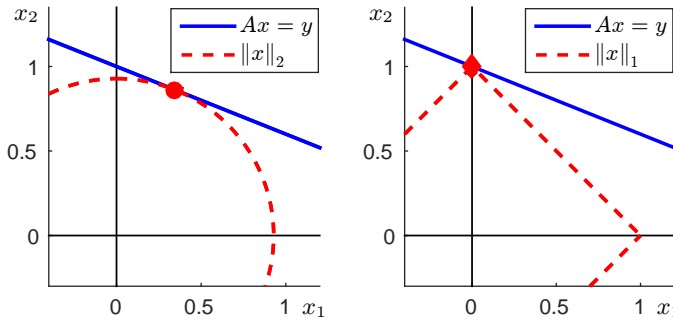

**Figure 1.** Illustration of 2-norm and 1-norm regularization for an underdetermined problem $\mathbf{A}\boldsymbol{x} = \boldsymbol{y}$ in $\mathbb{R}^2$. Generally, the minimum 1-norm solution (left) is zero in one of the two components. Such sparsity rarely happens when considering the minimum 2-norm solution (right). Similarly, the minimum 1-norm solution produces solutions with as many zero components as consistent with the data in higher dimensional underdetermined problems.

still produce a sparse approximation of the solution. Of course, the same is true for other penalty functions promoting certain properties of the estimate. Whether or not a signal is sparse is a matter of the representation system used. The solution might be non-sparse in the natural parameter space but have a sparse representation when transformed into a different space. To make use of the sparsity constraint, we present a representation system that allows for sparse representation for all possible solutions.

The field of signal and image processing offers a variety of transforms designed for sparse representation of oscillations, localized signals, edges and the like. Options include regular basis transforms, Fourier transforms, wavelets, shearlets, curvelets, among other options (see e.g. Diniz et al., 2010; Elad, 2010). However, it is not straightforward to find a sparsifying transform for a given application, and often a basis with its unique representation of the state is too restrictive. We consider a more flexible representation system called dictionary, described in detail in the next paragraph.

A dictionary is a collection of $N$ elementary functions $\boldsymbol{d_k} \in X$, called atoms, that can be combined linearly to represent the parameters, so $\boldsymbol{x} = \sum_{k=1}^{N} c_k \boldsymbol{d_k}$. These atoms can be thought of as building blocks of the signal. By the choice of the atoms, it can be ensured that there is at least one representation for each state $\boldsymbol{x} \in X$. Dictionaries are typically redundant representation systems, meaning that $N > \dim(X)$. This quality means that there are infinitely many representations $\boldsymbol{c}$ for the same parameter vector $\boldsymbol{x}$. We expect that at least one representation in a suitable dictionary is sparse, which means that true signal $\boldsymbol{x}^+$ is a

linear combination of only a few atoms. The sparse reconstruction approach, Eq. (6), can be used to select these atoms.

Consider an example for $X = \mathbb{R}^3$ with the dictionary

$$\mathbf{D} = (\boldsymbol{d_1}, \boldsymbol{d_2}, \boldsymbol{d_3}, \boldsymbol{d_4}) = \begin{pmatrix} \frac{1}{\sqrt{2}} & 1 & 0 & 0 \\ \frac{1}{\sqrt{2}} & 0 & 1 & 0 \\ 0 & 0 & 0 & 1 \end{pmatrix}.$$

Each column of $\mathbf{D}$ is an atom of norm one ($\|\boldsymbol{d_k}\|_2 = 1$). The vector $\boldsymbol{x} = (1,1,1)^t$ can be represented in the dictionary by coefficients $\boldsymbol{c} \in \mathbb{R}^4$ in infinitely many different ways as the dictionary is redundant. Some possible representations include

$$\begin{pmatrix} 0 \\ 1 \\ 1 \\ 1 \end{pmatrix}, \begin{pmatrix} -\sqrt{2} \\ 2 \\ 2 \\ 1 \end{pmatrix}, \begin{pmatrix} \sqrt{2} \\ 0 \\ 0 \\ 1 \end{pmatrix}, \begin{pmatrix} \frac{1}{\sqrt{2}} \\ \frac{1}{2} \\ \frac{1}{2} \\ 1 \end{pmatrix}, \dots$$

The first representation is a somewhat natural choice as it represents each dimension with a different atom. The third repre-
sentation has minimal 1-norm and the fourth minimal 2-norm. All other choices have more complicated structures. The third representation is also the sparsest possible representation. This example illustrates not only that the sparsest solution often coincides with the minimum 1-norm solution, but also how a redundant representation system is able to sparsify the signal with fewer nonzero entries than the vector it represents.

We assume that the estimated state in the AIM problem can be sparsely represented in a given dictionary $\mathbf{D}$, leading to the optimization problem

$$\boldsymbol{c}^* = \arg\min_{\boldsymbol{c} \in C} \frac{1}{2}\|\mathbf{L}_\delta(\mathbf{F}(\mathbf{D}\boldsymbol{c} + \boldsymbol{x_a}) - \boldsymbol{y_\delta})\|_2^2 + \alpha\|\boldsymbol{c}\|_1$$

$$\boldsymbol{x}^* = \mathbf{D}\boldsymbol{c}^*, \tag{7}$$

where $\boldsymbol{x_a}$ is an a priori estimate of the state. Again, the assumption when solving Eq. (7) is that the difference between true solution and a priori, $\boldsymbol{x}^+ - \boldsymbol{x_a}$, can be approximated by a linear combination of a small number of dictionary atoms $\boldsymbol{d_k}$. In contrast to the sparse reconstruction approach, Eq. (6), this assumption does not require that the flux field is sparse. A suitable dictionary provides a sparse approximation to many signals that are non-sparse in the standard representation system (see e.g. Candès et al., 2011; Starck et al., 2004; Elad, 2010). The approach is thus particularly well-suited to sparse problems, but it can also adeptly estimate non-sparse signals. We refer to this approach as sparse dictionary reconstruction.

### 3.4 Solving Tikhonov regularized inverse problems

The previous section formulates the AIM problem as an optimization problem using Tikhonov functionals. In the following paragraphs, we focus on efficient methods to solve problems (5), (6) and (7).

Henceforth, we only consider linear forward models $\mathbf{F}(\boldsymbol{x}) := \mathbf{A}\boldsymbol{x}$. Nonlinear forward models require additional properties for the existence of a minimizer and might have local minima. They are typically addressed by solving a sequence of linearized problems. Theory for nonlinear inverse problems is still an active field of study. Jin and Maass (2012) summarize the basic results.

For linear forward models the classical Tikhonov functional, Eq. (5), is strictly convex, and thus a unique global minimizer exists. The optimization problem can be solved by exerting the necessary conditions of first order, i.e. setting its derivative equal to zero. This setup leads to the linear equation

$$(\mathbf{A}^t\mathbf{L}_\delta^t\mathbf{L}_\delta\mathbf{A} + \alpha\mathbf{L_a^t}\mathbf{L_a})\boldsymbol{x} = \mathbf{A}^t\mathbf{L}_\delta^t\mathbf{L}_\delta\boldsymbol{y_\delta} + \alpha\mathbf{L_a^t}\mathbf{L_a}\boldsymbol{x_a}. \tag{8}$$

The solution to this equation is the minimizer of problem (5). A variety of methods exist to solve this linear equation. Our choice is a conjugate gradient method. Note the similarity of Eq. (8) to Bayesian inversion when $\mathbf{L}_\delta^t \mathbf{L}_\delta = \mathbf{R}^{-1}$ and $\alpha \mathbf{L}_a^t \mathbf{L}_a = \mathbf{Q}^{-1}$.

For the sparse reconstruction problem, Eq. (6), the functional to minimize is only convex and no longer differentiable everywhere. For this kind of problem, subgradient methods can be applied to find a minimizer. A fundamental contribution
was the Iterative Shrinkage Thresholding Algorithm (ISTA) (Daubechies et al., 2004), which is a simple iterative scheme consisting of a gradient and a shrinkage step:

$$\boldsymbol{x_{k+1}} = \mathbf{S}_{\alpha\beta}\left(\boldsymbol{x_k} - \beta \mathbf{A}^t \mathbf{L}_\delta^t \mathbf{L}_\delta (\mathbf{A}\boldsymbol{x_k} - \boldsymbol{y_\delta})\right)$$
$$\mathbf{S}_\lambda(\boldsymbol{x})_i := \max(|x_i| - \lambda, 0)\, \mathrm{sign}(x_i). \tag{9}$$

The stepsize $\beta$ must be chosen such that $0 < \beta < 2/\|\mathbf{L}_\delta \mathbf{A}\|^2$. The gradient step adds a non-sparse update to the current iterate.
Subsequently, the shrinkage operator $\mathbf{S}_\lambda : X \to X$ shrinks the updated parameters componentwise by $\lambda$ towards zero. This step ensures that only dominant components can increase to non-zero values. The algorithm converges rather slowly, but faster algorithms have been developed (Elad, 2010; Loris, 2009). We use the Fast Iterative Shrinkage Thresholding Algorithm (FISTA) (Beck and Teboulle, 2009).

The sparse dictionary reconstruction problem (7) translates into the sparse reconstruction problem (6) by defining $\tilde{\mathbf{A}} := \mathbf{A}\mathbf{D}$
and $\tilde{\boldsymbol{y}}_\delta := \boldsymbol{y_\delta} - \mathbf{A}\boldsymbol{x_a}$. This reformulation allows the use of the methods stated.

### 3.5   Bounds on the parameters

Some problems require a bound on the parameter space: $\boldsymbol{x} \in M \subset X$. Examples are nonnegative physical quantities like atmospheric mixing ratios. We only consider a nonnegativity constraint here, but the approach works for general closed convex subsets. When enforcing positivity, we use iterative methods to solve the optimization problems (5), (6) and (7) and couple the
update scheme $\mathbf{T}$ with a projection step $\mathbf{P_M}$ onto the set of permitted parameters $M$;

$$\boldsymbol{x_{k+1}} = \mathbf{P_M}(\mathbf{T}(\boldsymbol{x_k})).$$

For a positivity constraint, the projection is straightforward when the iteration is carried out in the state space $X$ by setting all negative parameters to zero. However, it can be complicated to translate these constraints to the corresponding space when sparsity is assumed in a different representation system (e.g. a dictionary). For our sparse dictionary reconstruction, we calculate
the corresponding parameters $\boldsymbol{x_k}$ to the current iterate $\boldsymbol{c_k}$ before projecting: $\boldsymbol{x_k} = \mathbf{D}\boldsymbol{c_k}$. Subsequently, the projection can be performed in parameter space, $\boldsymbol{x_k^+} = \mathbf{P_M}(\boldsymbol{x_k})$. Finally, we have to translate $\boldsymbol{x_k^+}$ back into dictionary space. Note, that there are infinitely many representations for the same state.

The iterative scheme of the sparse dictionary reconstruction creates a sparse representation $c_k$. Hence, one should also choose the sparsest representation for the projected state $\boldsymbol{x_k^+}$, which requires solving
$$\boldsymbol{c_{k+1}} = \arg\min_{\boldsymbol{c} \in C} \frac{1}{2}\|\mathbf{D}\boldsymbol{c} - \boldsymbol{x_k^+}\|_2^2 + \alpha\|\boldsymbol{c}\|_1. \tag{10}$$

The problem can be solved using the iterative shrinkage algorithm (see Eq. (9)). We can speed up the convergence with a good inital value, which is given by the current iterate $\boldsymbol{c_k}$. Moreover, the iteration does not need to run until convergence is reached

as the outcome will change in the next update step. Still, solving Eq. (10) for each iteration of the update scheme is a costly operation.

The projection step for the dictionary is difficult because the dictionary $\mathbf{D}$ is not invertible. We suggest the following heuristic approach; we select a subset of atoms from the dictionary that form a basis. The projection update is then only calculated for these components. This procedure might damage the sparsity of the current iterate. However, sparsity will be created by the next shrinkage step, if the update by the projection was not too large. It is important to note that this idea is heuristic, meaning that the algorithm may not converge against a minimizer of problem (7) restricted to nonnegative parameters $x$ in some cases.

## 3.6 Link to Bayesian inversion

The methods presented in this paper are formulated as Tikhonov regularizations. The inverse modeling community may be more familiar with the statistical formulation, namely Bayesian inverse modeling. In the following section, we briefly describe how both formulations overlap.

In a Bayesian inverse modeling setup, noise and unknown parameters are assumed to be realizations of known probability distributions. Given these distributions and the forward model, Bayes' theorem is used to infer the a posteriori distribution. The maximizer of the posterior probability density function, called maximum a posteriori solution, is often presented as a best estimate. Further evaluation of the posterior distribution also yields uncertainty bounds for the estimate.

We previously explained that covariance matrices for the noise or prior translate into weighting matrices for the norms in the Tikhonov formulation (see Sect. 3.2 and 3.3). For proper weighting matrices, Tikhonov regularization with 2-norm penalty as in Eq. (5) is equivalent to a Gaussian prior and a Gaussian noise model. By contrast, Tikhonov regularization with 1-norm penalty from Eq. (6) translates into a Laplacian prior and Gaussian noise. The probability density functions for Gaussian and Laplacian distributions are shown in Fig. 5. The Tikhonov approach only aims at the calculation of a best estimate, which compares to the maximum a posteriori solution in the Bayesian approach. Uncertainties can be assessed by additional calculations, which we present in Sect. 3.7.

Inversions with non-Gaussian priors, like the Laplacian in Eqs. (6) and (7), rarely have an analytical solution simplifying the calculation of the posterior distribution. The posterior distribution can also be approximated by samples created by Markov Chain Monte Carlo methods (e.g. Andrieu et al., 2003; Tarantola, 2005, Ch. 2). In those cases the computational cost for Bayesian inversion methods can become intractable. Tikhonov methods calculate a best estimate without further information about the underlying distribution. This property makes them more suitable for computationally demanding nonlinear or large scale problems if an uncertainty analysis is not required.

## 3.7 Error analyis

To judge the quality of an estimate, it is necessary to know the uncertainty associated with each estimated parameter. For Bayesian methods, these uncertainties and the best estimate are deduced from samples of the posterior distribution if no analytical expressions exist. For the Tikhonov methods used in this work, uncertainty estimates are an extra calculation performed

after the retrieval of a best estimate. In this section, we present an uncertainty analysis for Tikhonov methods based on Rodgers (2000, ch. 3).

We call the true parameters $x^+$ and the a priori $x_a$. Let $\mathbf{R}$ denote a general inversion method, $\mathbf{R} : Y \to X$, and $\mathbf{F}$ a general forward model, $\mathbf{F} : X \to Y$. Then, the best estimate is given by

$$5 \quad x^* = x_a + \mathbf{R}(y_\delta - \mathbf{F}(x_a)) = x_a + \mathbf{R}(\mathbf{F}(x^+) - \mathbf{F}(x_a) + \delta).$$

We linearize the forward model and reconstruction method to determine the first order terms

$$x^* = x_a + \frac{\partial \mathbf{R}}{\partial \mathbf{y}} \frac{\partial \mathbf{F}}{\partial \mathbf{x}} (x^+ - x_a) + \frac{\partial \mathbf{R}}{\partial \mathbf{y}} \delta. \tag{11}$$

For linear forward models we have $\frac{\partial \mathbf{F}}{\partial \mathbf{x}} = \mathbf{A}$, but the reconstruction methods remain nonlinear. Thus the analysis depends on the point of linearization. The total error can be differentiated between smoothing and (total) measurement error:

$$10 \quad x^* - x^+ = \underbrace{\left( \frac{\partial \mathbf{R}}{\partial \mathbf{y}} \frac{\partial \mathbf{F}}{\partial \mathbf{x}} - \mathbf{I} \right) (x^+ - x_a)}_{\text{smoothing error}} + \underbrace{\frac{\partial \mathbf{R}}{\partial \mathbf{y}} \delta}_{\text{measurement error}}. \tag{12}$$

The measurement error describes how noise on the measurement data propagates to errors in the estimated parameters. Recall that the definition of noise used here includes errors in the measurement, the forward model and numerical approximations. Reconstruction methods try to suppress the effect of noise on the parameters by stabilizing the unstable inversion (e.g. via a penalty term or an a priori). This modification introduces the smoothing error. For ill-posed problems, a smaller smoothing error results in a greater measurement error and vice versa. The smallest total error is expected when both terms are approximately balanced (see e.g. Hansen, 2010, Ch. 5).

The exact total, smoothing and measurement error can be calculated from the true solution, $x^+$, the estimate under noisy data, $x^*$, and the estimate to noiseless data using the same regularization parameter as in the noisy case. The errors are given by

$$20 \quad \underbrace{x^* - x^+}_{\text{total error}} = \underbrace{\boldsymbol{R}_{\alpha^*}(y_\delta - \mathbf{F}(x_a)) - \mathbf{R}_{\alpha^*}(y - \mathbf{F}(x_a))}_{\text{measurement error}} + \underbrace{x_a + \mathbf{R}_{\alpha^*}(y - \mathbf{F}(x_a)) - x^+}_{\text{smoothing error}}. \tag{13}$$

Note that this equation does not require a sensitivity matrix.

In real data problems, the error terms in Eqs. (12) and (13) are impossible to calculate as they require knowledge of the true solution $x^+$ as well as the noise $\delta$. Instead, it is common to estimate reliable uncertainties that bound the actual errors.

To find such bounds for the smoothing error, Bayesian methods make additional assumptions on the true solution by applying so called a priori knowledge. Comparable source conditions also exist for Tikhonov methods (e.g. Natterer, 1984; Tautenhahn, 1998; Engl et al., 1989; Jin and Maass, 2012), but such assumptions are often hard to guarantee.

Without applying a priori knowledge, the best one can do is to analyze the sensitivity matrix $\mathbf{S} := \frac{\partial \mathbf{R}}{\partial \mathbf{y}} \frac{\partial \mathbf{F}}{\partial \mathbf{x}}$, sometimes called averaging kernel matrix. For linear forward models $\mathbf{A}$ it can be approximated by

$$\mathbf{S}_{:,\mathbf{k}} := \frac{\mathbf{R}(y_\delta - \mathbf{A}x_a + \mathbf{A}\Delta x_k) - \mathbf{R}(y_\delta - \mathbf{A}x_a)}{\|\Delta x_k\|}. \tag{14}$$

The $k$-th column of this matrix expresses how the estimate reacts on a perturbation in the $k$-th parameter of the true fluxes, $\Delta x_k$. Its structure gives an insight into the smoothing error. For an ideal sensitivity matrix equal to the identity the smoothing error vanishes (see Eq. (12)). Thus, the closer the sensitivity matrix is to the identity, the smaller smoothing error can be expected. For nonlinear reconstruction methods $\mathbf{R}$, interpretation of the sensitivity matrix is difficult. The information is only local and cannot predict reactions of the estimate for perturbations different from $\Delta x_k$ via Eq. (12). Even the normalization in Eq. (14) might be misleading, if the amplitude of the perturbation $\|\Delta x_k\|$ is not provided. However, sensitivities are still meaningful for assessing the accuracy of the estimate. We describe further details on how we analyze the sensitivity matrix in Sect. 4.6. The numerical computation requires one to solve a reconstruction problem per parameter, which can be done in parallel, but might still be infeasible for large scale problems.

Even in real data problems, one often has access to the noise characteristics. Uncertainty bounds for the measurement error can be approximated via resampling of the noise and recalculation of the estimate under this noise for a sufficient number of samples. The distribution of estimates to different realizations of the noise yields the uncertainties commonly expressed by standard deviations. If the noise characteristics are unknown, resampling can be achieved by bootstrapping of the residual of the estimate (see Banks et al., 2010). This numerical approach is computationally demanding, but can be run in parallel.

# 4   Case study: Methane emissions in the United States

We apply the sparse dictionary reconstruction method in an atmospheric inverse modeling setup. We use anthropogenic methane emissions in the United states as a synthetic case study, the same case study used in Miller et al. (2014). This setup provides an opportunity to compare flux estimates obtained using different methods against the known, synthetic fluxes. This section describes the details of the case study and the methods used. Before we specify how the sparse dictionary reconstruction method is set up, we compare Tikhonov regularization with 2-norm and 1-norm penalty to visualize the effect of the 1-norm.

## 4.1   Case study details

We estimate emissions for the North American mainland ($25 - 55°\ N$ and $145 - 51°\ W$) on a $1°$ by $1°$ grid (land grid cells only). We use a combination of synthetic in-situ aircraft and tall tower measurements, that were available during May to September 2008 from operations by the NOAA Earth Systems Research Laboratory (NOAA; Andrews et al., 2014), the United States Department of Energy (Biraud et al., 2013) and the START08 aircraft campaign (Pan et al., 2010). Footprints for these measurements, that define the forward model, are calculated using the WRF-STILT model (see Sect. 2).

Synthetic methane emissions are generated from the Emission Database for Global Atmospheric Research (EDGAR). We project anthropogenic methane emissions from the EDGAR v3.2 FT2000 inventory (Olivier and Peters, 2005) onto our model grid. These emissions are constant in time during our observation period. As discussed in Miller et al. (2014), newer versions of the EDGAR inventory are available, but we use this version for reasons of comparison to the previous study. Moreover, we work with simulated data only, so there is no strict need to use the most recent inventory version. Rather the solution should comprise typical features, which we assume holds for this version as well.

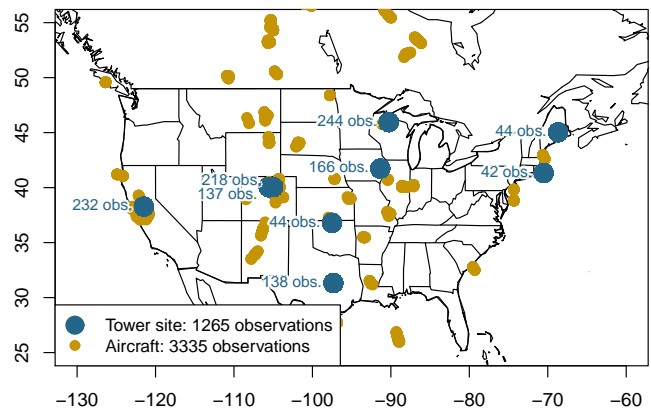

**Figure 2.** Available in-situ methane measurements for May to September 2008

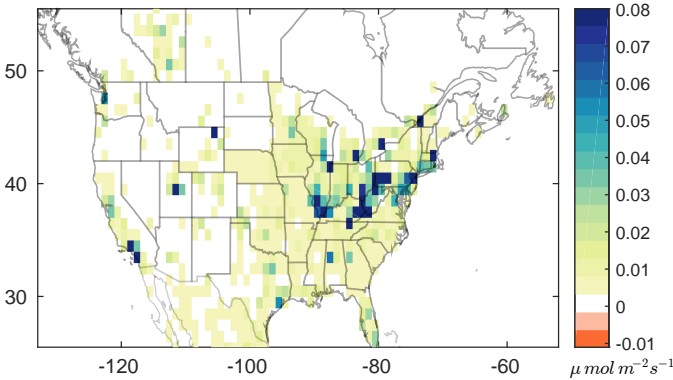

**Figure 3.** US methane emissions from the EDGAR v3.2 FT2000 in the native $1°$ by $1°$ resolution. The largest source regions, New York and Eastern Kentucky, have fluxes higher than three times the limit of the colormap.

The simulated noisy measurements are calculated by applying the linear WRF-STILT forward model to the EDGAR fluxes and adding Gaussian noise of realistic magnitude. The noise vector is sampled from the multivariate Gaussian distribution with a diagonal covariance matrix. Miller et al. (2013) estimte the values of this matrix using real observations (Fig. 2) and restricted maximum likelihood.

### 4.2 Classical Tikhonov regularization vs. sparse reconstruction

The fluxes are temporally constant in the inversion setup here, so each of the 1469 land grid cells has only one unknown emission parameter ($\boldsymbol{x} \in \mathbb{R}^{1469}$). On the other hand, there are 4600 total measurements, so $\boldsymbol{y_\delta} \in \mathbb{R}^{4600}$. Despite the fact that there are more measurements than unknowns, the inverse problem is still ill-posed as many measurements yield similar pieces of information. We estimate the surface fluxes $x$ only by knowledge of the forward model $\mathbf{A}$ defined by the footprints, the noisy measurements $\boldsymbol{y_\delta}$ and the noise characteristics determined by the noise covariance matrix $\mathbf{R}$. In this scenario $\mathbf{R}$ does not have

any off-diagonal entries, thus it is easy to calculate $\mathbf{L}_\delta$ from $\mathbf{L}_\delta^t \mathbf{L}_\delta = \mathbf{R}^{-1}$. In general cases we recommend using the Cholesky factorization. For all problems, we use a zero a priori, so $\boldsymbol{x_a} = \mathbf{0}$. More advanced a priori models should be considered in real data scenarios.

We start by comparing Tikhonov regularization with classical 2-norm penalty

$$\boldsymbol{x}^* = \arg\min_{\boldsymbol{x} \in \mathbb{R}^n} \frac{1}{2} \|\mathbf{L}_\delta(\mathbf{A}\boldsymbol{x} - \boldsymbol{y_\delta})\|_2^2 + \frac{\alpha^*}{2}\|\boldsymbol{x}\|_2^2 \tag{L2}$$

and Tikhonov regularization with sparsity constraint

$$\boldsymbol{x}^* = \arg\min_{\boldsymbol{x} \in \mathbb{R}^n} \frac{1}{2} \|\mathbf{L}_\delta(\mathbf{A}\boldsymbol{x} - \boldsymbol{y_\delta})\|_2^2 + \alpha^*\|\boldsymbol{x}\|_1. \tag{L1}$$

The optimal regularization parameter $\alpha^*$ is approximated by Morozov's discrepancy principle for each problem; we start with a value $\alpha_0$ that is certainly too large, so the corresponding minimizer is the a priori $\boldsymbol{x_a}$, here $\boldsymbol{x_a} = \mathbf{0}$. Then, the regularization parameter is reduced iteratively by $\alpha_k = q\alpha_{k-1}$ with $0 < q < 1$ and the corresponding minimizer $\boldsymbol{x}_{\alpha_k}$ is calculated. For each minimizer we check whether

$$\|\mathbf{L}_\delta(\mathbf{A}\boldsymbol{x}_{\alpha_k} - \boldsymbol{y_\delta})\|_2 < \tau\bar{\delta}, \tag{15}$$

where $\bar{\delta} = \|\delta\|_2$ is the expected noise-level and $\tau > 1$. If it holds, an appropriate regularization parameter $\alpha^* := \alpha_k$ is found and the corresponding minimizer $\boldsymbol{x}_{\alpha^*}$ is our best estimate $\boldsymbol{x}^*$. In Eq. (15), we have $\bar{\delta} = \sqrt{m}$, $m = 4600$ as the noise is normalized by $\mathbf{L}_\delta$.

For a fixed $\alpha$-value, we use a conjugate gradient method to solve problem (L2) via Eq. (8). We include a speed-up by Frommer and Maass (1999), which detects in early iterations whether or not Morozov's criteria, Eq. (15), can be reached for the given parameter. This speed-up allows one to continue with the subsequent smaller regularization parameter $\alpha$ before convergence is reached.

We use FISTA, which is an accelerated version of ISTA (see Sect. 3.4), to determine the sparse reconstruction solution, Eq. (L1), to a fixed $\alpha$-value. It uses a weighted combination of the last two iterates to calculate an update instead of using the last iteration only.

## 4.3 Preliminary results: Classical Tikhonov regularization vs. sparse reconstruction

Figure 4 shows the methane emission estimates by the Tikhonov methods L2 and L1. Small sinks appear in both estimates because both methods are not restricted to positive emissions. It is worth noting that the measurements include some negative values, due to the fact that they are enhancements above a background level and are perturbed by noise. Even without negative observations, the estimated emissions may include negative values, if not explicitly enforced. An overestimation in one grid cell and an underestimation in another might still be consistent with the data because the data are not sufficient to fully constrain all flux parameters in an ill-posed inverse problem.

The estimates differ (Fig. 4 but explain the data up to the noise. Most inverse problems are under-constrained by the data. As a result, the penalty term has a large effect on the final estimate and explains many of the differences between the L1 and

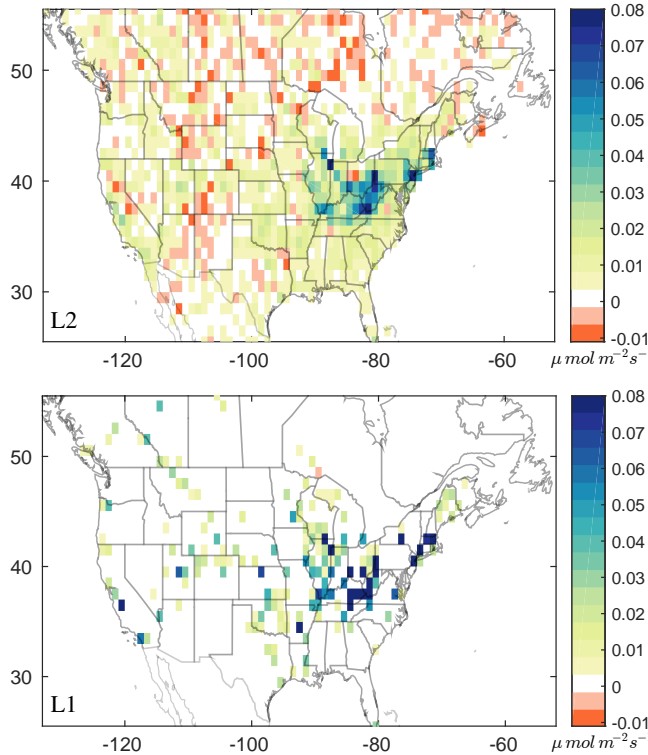

**Figure 4.** Emission estimates using Tikhonov regularization with classical 2-norm penalty (see Eq. (L2)) and sparsifying 1-norm penalty (see Eq. (L1)) inverted from noisy simulated methane measurements. The true flux field is shown in Fig. 3. While L2 shows the typical smoothing effect, L1 concentrates the signal, which results in better estimates of large sources, but also tends to explain regional emissions by larger point sources.

L2 estimates. As expected, L2 produces an emission field that is smooth. Large sources are avoided to minimize the 2-norm. In contrast, L1 produces emissions that are larger in magnitude but more concentrated to few pixels. The resulting estimated emission field is sparse.

Often, the sparse emission field better estimates large sources such as those from major cities (see e.g. Salt Lake City emissions at $111°$ W, $40°$ N). Such large isolated pixel emissions are smeared to regional emissions by L2. However, L1 misplaces some of these large emitters (see e.g. the San Franciscan Bay Area). This particular misplacement is present in all methods and is caused by a combination of small footprint information and the measurement noise. The source is placed to the correct grid cell for other realizations of the noise. Also, L1 estimates regional emissions such as those in Kansas or Arkansas falsely as large pixel emitters and neglects many small sources. The estimate given by L2 much better reconstructs these regional emissions. A grid cell by grid cell comparison favors the L2 estimate. L2 also prodoces a better estimate for the total emissions (see Table 1).

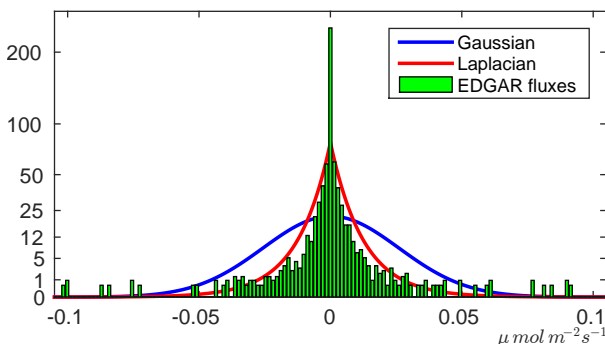

**Figure 5.** Normalized histogram of randomly signed EDGAR fluxes. The histogram data has been used to estimate the parameters of corresponding Gaussian and Laplacian probability density functions. Note that only ten grid cells have emissions larger than $0.1\,\mu\,\mathrm{mol}\,\mathrm{m}^{-2}\,\mathrm{s}^{-1}$ with the largest one reaching $0.29\,\mu\,\mathrm{mol}\,\mathrm{m}^{-2}\,\mathrm{s}^{-1}$. Even though the center bin is largely populated, only less than $10\%$ of all fluxes are equal to zero.

The histogram of the EDGAR fluxes in Fig. 5 supports the use of the sparsity constraint. We assign a random sign to each flux as fluxes are nonnegative, in contrast to the a priori models of L2 and L1. The resulting empirical distribution agrees much better with a Laplacian than with a Gaussian prior. Recall that L1 corresponds to a Laplacian and L2 corresponds to a Gaussian prior.

Based on these preliminary results we conclude that the estimate using L2 is closer to the true EDGAR emissions than L1, but the estimate is not satisfying for the reconstruction of large emitters due to the smoothing effect. Also, this methane emissions case study is not suitable for sparse reconstruction in the standard representation system.

## 4.4 Sparse dictionary reconstruction

The preliminary results in the previous section show that the classical 2-norm regularization estimates a flux field that is too
smooth. Large pixel sources such as those from cities are smoothed out. Sparse reconstruction improves the estimate of these large sources. However, we also observe that regional sources are likely incorrectly represented as point sources and that the total emissions are underestimated. The EDGAR solution is neither smooth nor naturally sparse because we expect methane emissions of differing sizes in most grid cells. We seek to find a representation system that is able to sparsely represent such emission fields.

We employ a dictionary to achieve this goal. We therefore need to select atoms, such that the dictionary can sparsely approximate all methane emission patterns. Efficient dictionaries can be created using learning algorithms, but a set of training data is required (see e.g Mairal et al., 2014). We could extract training data from the EDGAR inventory for other regions, learn a dictionary and use it for the United States setup, but results could be too optimistic as this will not be an option for real data scenarios. Our approach is to identify typical source shapes and include these shape functions as atoms in our dictionary. The
sparse reconstruction approach will then select those atoms that explain the data in the sparsest way.

The 1° by 1° model grid is too coarse to identify individual sources. Many typical methane sources such as cities, landfills and waste, industrial facilities and mining do not extend beyond one grid cell. To represent these grid cell emissions efficiently, we include the pixel basis in our dictionary. Metropolitan areas, livestock areas, oil and gas fields might extend over several pixels though. Thus, we also add circular peak shape functions (see Fig. 6). We could add more functions with bigger and more

complicated shapes. That approach not only requires more computational time, but it also leads to more redundancy. It is our intention to create some redundancy but only to the degree that it helps sparsify the representation of our possible emission fields. From numerical experiments, we find that including bigger shape functions do not add value to the reconstruction.

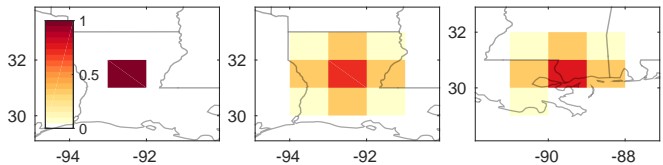

**Figure 6.** A selection of atoms from the dictionary used for the sparse dictionary reconstruction method. These atoms are scaled to represent the state vector via linear combination. The left and the middle element are the basic shapes centered in each grid cell of the domain. At coasts and lakes these shapes are limited to land grids. All atoms are normalized in the 2-norm. The dictionary chosen here also holds a constant background function.

Another option to sparsify the representation is to use atoms that cover a large portion of the domain. A background is best represented by a constant function. With the same argument we could add regional background functions. We in fact find that

a division into regions as shown in Fig. 10 would improve the estimate, but the placement of those regions is partly inspired by looking at the true EDGAR fluxes.

Michalak et al. (2004) presents a geostatistical inversion as an extension of L2. This approaches uses a model of the mean in place of a traditional prior emissions estimate. It spatially correlates regions based on geostatistical information such as population density or agricultural use. This setup results in shape functions, which we could include in our dictionary as well.

The difference between both approaches is that our coefficients receive penalization to select the atoms during the inversion, whereas the shape functions are preselected and unconstrained in the geostatistical inversion. Weighting the penalty on the coefficients individually would translate from one approach to the other.

For our experiments, we decide not to include atoms that are constructed from EDGAR or geostatistical data. We use a pixel basis, a basis with peaks that extend into the direct neighbors (see Fig. 6) and a background function for the entire domain.

Thus, we have $\mathbf{D} \in \mathbb{R}^{n \times N}$ with $N \approx 2n+1$. All atoms are normalized in the 2-norm. Generally speaking, our dictionary holds functions to represent processes at different spatial scales.

To estimate the flux parameters $x$ with sparse dictionary reconstruction we solve

$$c^* = \arg \min_{c \in \mathbb{R}^N} \frac{1}{2} \|\mathbf{L}_\delta(\mathbf{A}(\mathbf{D}c + x_a) - y_\delta\|_2^2 + \alpha^* \|c\|_1$$

$$x^* = \mathbf{D}c^*. \tag{L1 DIC}$$

As before, we use a zero a priori, $x_a = 0$, the discrepancy principle, Eq. (15), to determine the optimal regularization parameter $\alpha^*$ and FISTA to solve problem (L1 DIC) for a given $\alpha$. A pseudocode is included in the supplementary information.

## 4.5 Enforcing positive fluxes

For further analysis, we add a positivity constraint on the flux parameters, $x \in R_+^n$, which we denote by the suffix POS. We use the projection approach described in Sect. 3.5 for all methods to enforce positivity. Projecting the iterates of a conjugate gradient method may lead to poor performance, as the special structure of the search directions is lost. Thus, we also use FISTA with a shrinkage operator for the 2-norm penalty when solving L2 POS. Note that the projection step for L1 DIC POS is more demanding as it involves the transition from parameter to dictionary space (see Eq. (10)). Instead, we apply the suggested heuristic nonnegativity update (see Sect. 3.5) using the pixel basis as an invertible submatrix to correct for negative fluxes. Further details are included in the supplementary information.

By enforcing positive parameters, three different constraints determine the final estimate: positivity, data and minimal norm. Often, these constraints may pull the estimate in different directions. The final estimate depends on the balance between them. Using a projection, positivity is always enforced. The emissions will also explain the given data up to the noise as long as Morozov's discrepancy principle is fulfilled. The most flexible constraint in this setup is thus the smoothness or sparsity assumption defined through the penalty term because it is the most uncertain of all constraints.

## 4.6 Error analysis

We carry out two types of analysis to measure the quality of the estimates. First, we perform an uncertainty analysis based on knowledge about the noise characteristics but without knowledge about the true fluxes as would be the case for many real data scenarios. As discussed in Sect. 3.7, we assess smoothing and measurement error separately by analyzing the sensitivity matrix and by resampling of the noise respectively. In a second analysis, we make use of the true EDGAR solution and calculate the exact total, smoothing and measurement errors.

The smoothing error describes the error that results from stabilizing the inversion. It can only be estimated if additional assumptions on the true fluxes are made. Without such assumptions, the best one can do is to analyze the sensitivity matrix (see Eq. (14)). As mentioned before, an ideal sensitivity matrix is equal to the identity. We address two measures: the column sum and the diagonal. The column sum of the sensitivity matrix should be close to one. Otherwise, the method over- ($> 1$) or under estimates ($< 1$) in that region. The diagonal of the sensitivity matrix shows the sensitivity of the parameter that is perturbed. Values close to one indicate high confidence in the reconstruction. Smaller values are either a consequence of smoothing or of not being sensitive at all. The latter is captured by looking at the column sum as well.

The measurement error shows the influence of the noise on the estimated parameters. We estimate uncertainty bounds for the measurement error based on 1000 samples of noise as described in Sect. 3.7. We express the uncertainties by two standard deviations of the empirical distribution of estimates corresponding to these samples. The exact total, smoothing and measurement error are calculated using Eq. (13).

## 4.7 Comparison to other methods

We compare our approaches to state-of-the-art methods studied in Miller et al. (2014). The scope that article is to analyze different formulations to enforce positive parameters. The methods are:

– Standard inversion: This is a geostatistical approach following Michalak et al. (2004). It does not include a positivity constraint and is taken as a benchmark method in Miller et al. (2014).

– Transform inversion: Flux parameters are enforced to be positive by a power transformation (see Snodgrass and Kitanidis, 1997). This technique can also be used to straighten skewed parameter distributions.

– Lagrange multiplier method: Positivity is enforced by formulating an optimization problem with an inequality constraint, which is solved via the Lagrangian function. As a deterministic method, no direct uncertainty estimates are given, but they can be approximated using the approaches from Sect. 3.7.

– Gibbs sampler: The Gibbs sampler belongs to the group of Markov Chain Monte Carlo (MCMC) methods. These methods can generate realizations of complicated probability distributions such as the posterior distribution to non-Gaussian priors in a Bayesian inversion framework. MCMC methods differ in the way these realizations are calculated. One can estimate statistical quantities such as mean and standard deviation given a sufficient number of realizations. Positivity is formulated in the prior distribution. In theory, one could implement one of several MCMC algorithms (see Miller et al., 2014), but we focus on the Gibbs sampler here (e.g. Michalak and Kitanidis, 2003).

All methods have been discussed in a Bayesian inversion framework. Further details and references are given in Miller et al. (2014).

## 5 Results

In this section, we analyze the performance of our suggested sparse dictionary reconstruction method L1 DIC POS in the AIM scenario described in the previous section. First, we compare it with the methods L2 POS and L1 POS and carry out an uncertainty analysis and an analyis of the exact errors. Then, we include the methods from Miller et al. (2014) into the comparison. Finally, we analyze the ability of each method to reproduce spatially discrete emissions from oil and gas extraction in the Barnett Shale region of Texas.

## 5.1 Methane emission estimates

Figure 7 shows the estimated emissions using L2 POS, L1 POS and L1 DIC POS. Our first observation is that there are no sinks as positive fluxes are enforced by all methods. The results for L2 POS and L1 POS are close to the ones obtained by L2 and L1 (see Fig. 4 and Sect. 4.3), setting negative fluxes to zero. However, projecting the final estimate (see Sect. 3.5) should be avoided as the mismatch between modeled and measured data increases and thus does not make full use of the information

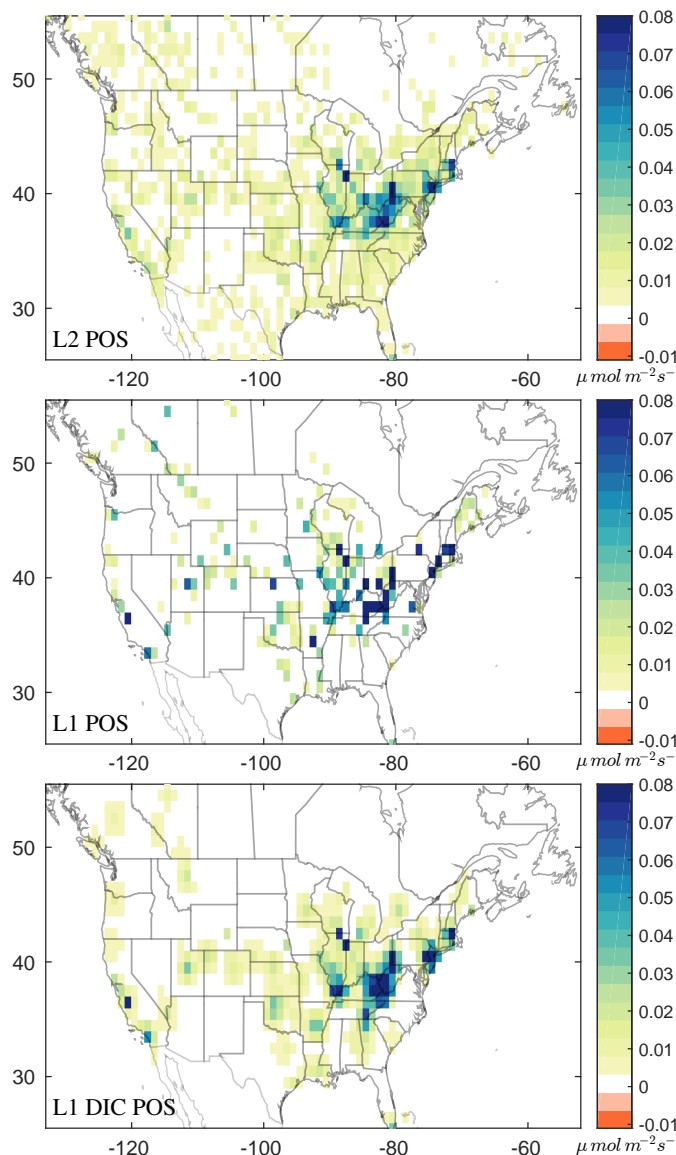

**Figure 7.** Emission estimates from the methods L2 POS, L1 POS and L1 DIC POS inverted from noisy simulated methane measurements. The true flux field is shown in figure 3.

in the data. A projection step in every iteration allows the algorithm to correct for this problem in the next update. We measure significant improvement excluding sinks, especially for L2 (see Table 1).

In contrast to L1 POS, the solution of L1 DIC POS does not look sparse, as sparsity is enforced on the coefficients of the dictionary. The background function in the dictionary is selected to represent a base level of small emissions (not visible in the color map). It improves the inversion's ability to accurately estimate total US emissions. Regionally, other atoms are added

and subtracted from this background level. The smooth character of the estimate in many regions is a result of the broader dictionary functions (see Fig. 6). The method adds pixel sources where local hot spots are assumed.

L1 DIC POS shows significant improvement in the estimate of localized sources against L2 POS (e.g. when looking at West Coast emissions or Salt Lake City) but a slight setback against L1 POS. Sometimes, these emission peaks might be misplaced (e.g. San Francisco Bay Area). Regions of significant emissions like in the Midwest are often reasonably well reconstructed, but the method still tends to spatially concentrate these sources. This localization property is a consequence of the sparsity constraint because the flux field is represented by as few atoms as possible.

We observe that the locations of significant sources agree much better with both sparsity methods L1 POS and L1 DIC POS than with the classical L2 approach. This result can be explained by the fact that the sparse schemes look for the dominant sources. Even if the magnitude is not captured exactly, the method L1 might be used in applications to identify the center of source locations.

## 5.2  Sensitivity, measurement uncertainty and error analysis

As described in Sects. 3.7 and 4.6 we assess smoothing and measurement error separately. In a first analysis, we ignore the known, true emission field and analyze the sensitivity matrix and the measurement uncertainties as if we faced a real data problem. In the error analysis, we study the exact total, smoothing and measurement errors.

### 5.2.1  Sensitivity analysis

The sensitivity matrix gives the best insight to the smoothing error without knowledge of the true fluxes. Each column describes, how an additional pixel source would change the flux estimate. A perfect sensitivity matrix is thus equal to the identity. The column sum indicates regions that are over- or underestimated. This phenomenon can only be observed in regions with small footprint information outside the main study area, namely Florida, Mexico and Central and Eastern Canada. The locations are similar for all methods, but L1 POS is far more biased in those regions. Table 1 shows that L1 POS indeed poorly estimates the total emissions.

The most valuable information is contained on the diagonal of the sensitivity matrix, plotted in Fig. 8 (left). The diagonal shows the sensitivity of the parameter that is perturbed. Unsurprisingly, all methods are most sensitive in the vicinity upwind of tower observation sites. Also, we observe that both sparse reconstruction schemes are less sensitive than L2 in regions that are poorly constrained by the data and have an increased sensitivity in regions of greater footprint values. Large sensitivities can also be found where parameters are active, i.e. in grid cells with nonzero emission estimate.

The interpretation of the senstivity matrix is slightly different for both types of methods. If we excluded the positivity constraint, L2 (POS) would be a linear method, meaning that the sensitivity matrix is independent of the parameters and could be used to predict how additional sources would be reconstructed using Eq. (11). Nonlinearity due to the positivity constraint should have little influence for positive parameters. In contrast, the sparsity constraint adds to the nonlinearity. As a result, the sensitivity matrix may be different for each flux field. The sensitivity for small sources may be small or even zero in some regions, but large sources or sources in several neighboring grid cells could still be reconstructed. We use a rather large

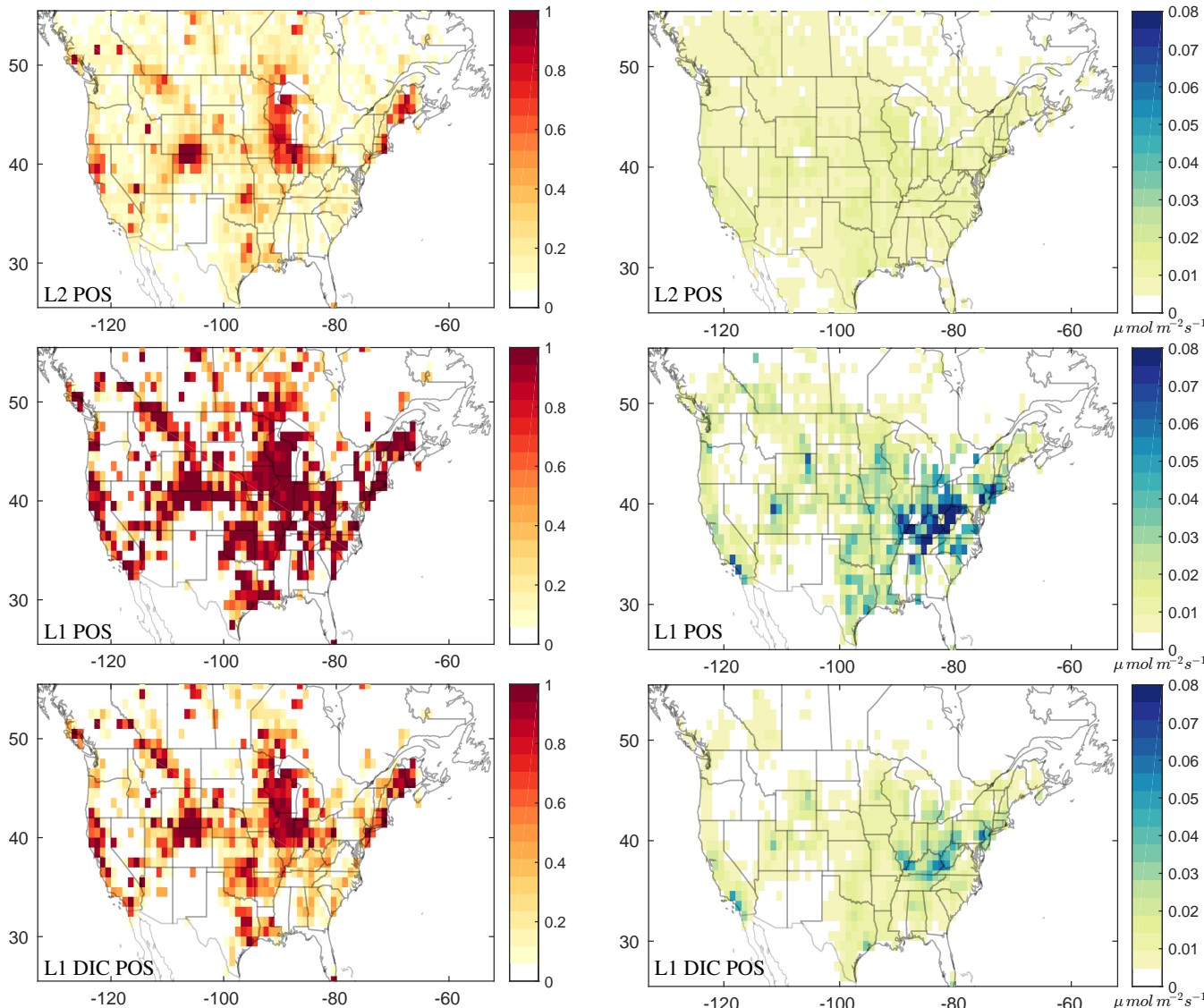

**Figure 8.** Left: Diagonal of the numerically calculated sensitivity matrices for large deviations of $\|\boldsymbol{\Delta x_k}\| = 0.1\mu$ mol m$^{-2}$ s$^{-1}$. Right: Two standard deviation uncertainties due to noise on the measurements.

perturbation to approximate the sensitivity matrices. Therefore, Fig. 8 (left) shows the fraction of a single large pixel source that is reconstructed at a given location.

### 5.2.2 Measurement uncertainty analysis

The measurement uncertainties describe the uncertainties in the estimate from the noise. We approximate these uncertainties
5   by resampling the noise. Two standard deviations are plotted in Fig. 8 (right). The reason for regularization is to limit the

influence of noise on the estimate. However, no influence at all would also mean that the method is not sensitive to data. For the L2 POS approach measurement, uncertainties are rather equally distributed across the full domain excluding the poorly constrained regions of Mexico and Canada.

The sparse reconstruction method L1 POS has much larger measurement uncertainties, particularly in places where large emitters are estimated. On the other hand, the estimated uncertainties are small or even zero in regions where the sensitivity is small. This is a consequence of the thresholding algorithm. The method reacts with its active, nonzero parameters on small perturbations in the data. The set of these active parameters is only adapted by significant changes. Theoretically, uncertainties can be equal to zero but it is likely that zero uncertainties are a result of a limited number of samples used for their calculation. It is important to keep in mind that these uncertainties only represent the effect from noise on the estimated parameters. Uncertainties for the smoothing error will be larger where the sensitivity is small.

L1 DIC POS looks more robust to noise than L1 POS. Similarly to L1 POS, there are large areas with neglectable measurement uncertainties. The uncertainty correlates with the magnitude of the estimated emissions.

### 5.2.3   Error analysis

Figure 9 displays the smoothing and measurement errors. Both errors are calculated using Eq. (13) for the methods L2 POS, L1 POS and L1 DIC POS. The errors shown here are linked to this particular realization of noise. The mean squared norm for the errors are calculated for comparison.

It is important to point out that it is misleading to look at either the smoothing or measurement error without the other. For ill-posed inverse problems, a small smoothing error comes at the expense of a larger measurement error and vice versa. A well chosen regularization parameter balances both errors such that the total error is minimized. For our methods, we observe that the measurement error is always smaller but the ratio is different for each method.

The measurement noise causes deviations on the estimated coefficients. This effect is described by the measurement error. For L2 POS, these deviations affect most parameters whereas for L1 POS the effect is larger but mostly limited to the active nonzero parameters. For L1 DIC POS, the noise affects the active atoms of the dictionary.

The smoothing error results from the stabilizing effect of the reconstruction methods. Because L2 POS aims at smooth emission fields, large pixel emissions are generally a combination of underestimation in that particular grid cell and overestimation in the vicinity. This smoothing effect does not manifest in the L1 POS (e.g. for the Salt Lake City emission). For L1 DIC POS smoothing can happen when spatially larger dictionary elements are selected.

While over- and underestimation are approximately equal for the measurement error, the smoothing error indicates whether a method over- or underestimates in general. Because of the zero a priori, all methods are expected to underestimate, but only L1 POS significantly underestimates (see also Table 1).

L1 POS reduces the smoothing effect in some locations, but smoothing and measurement error are larger than for the other methods because too many small sources are suppressed. Sparse dictionary reconstruction has significantly less smoothing error and thus gives the best estimate of the EDGAR fluxes. These results do not rule out the L1 POS method in general, but

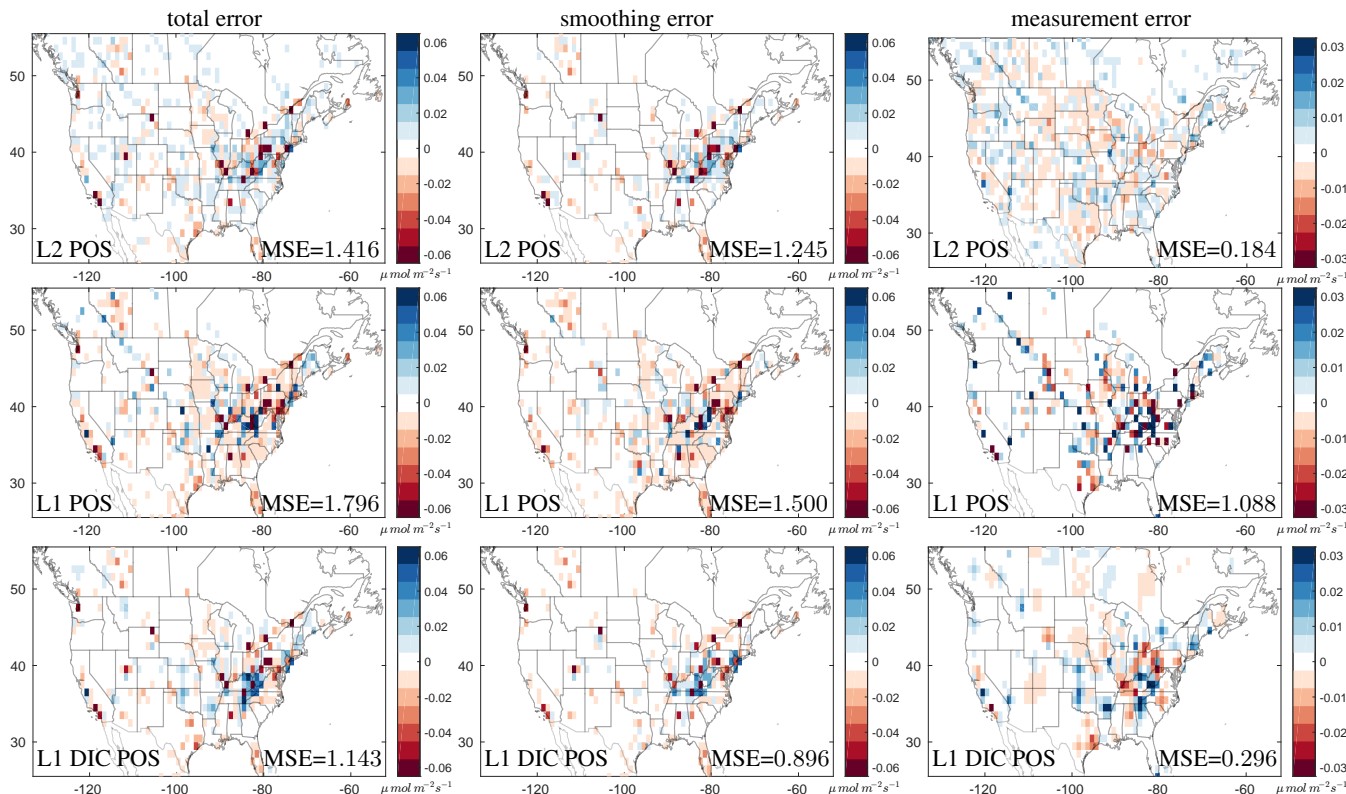

**Figure 9.** Total (left), smoothing (middle) and measurement error (right) for the methods L2 POS, L1 POS and L1 DIC POS. Underestimation is colored in red, whereas blue colors represent overestimation. The mean squared error (MSE) calculated for each panel is given in $10^{-4} \left( \mu \, mol \, m^{-2} s^{-1} \right)^2$. Note that the color coding is different for the measurement error.

it suggests that this particular case study is not naturally sparse. However, the dictionary representation is able to sparsify the signal and is thus well suited for these types of problems.

## 5.3 Comparison to other methods

In this section, we evaluate the estimates of our Tikhonov based methods by comparing them to the estimates of the methods
5   studied in Miller et al. (2014) (see Sect. 4.7).

### 5.3.1 General results

We examine the reconstruction quality using several measures, each of which focuses on aspects or qualities. These measures are:

$$\text{relative total error} := \frac{\int_\Omega x^+ - \int_\Omega x^*}{\int_\Omega x^+} \tag{16}$$

$$\text{relative regional error} := \frac{\int_\Omega |x^+ * \bigcirc - x^* * \bigcirc|}{\int_\Omega (x^+ * \bigcirc)} \tag{17}$$

$$\text{relative local error} := \frac{\int_\Omega |x^+ - x^*|}{\int_\Omega x^+}. \tag{18}$$

The local error compares the estimated and known, synthetic fluxes within each individual grid cell while the total error sums up all grid cell emissions to a North American flux before comparison. We choose to include the regional error as an intermediate measure between those two. The idea is to see if the total emissions over a region around the grid cell is estimated correctly. This approach relativizes the smoothing effect of the reconstruction methods. Here, $\bigcirc$ is a circular filter that gives a weighted sum of the neighbouring grid cells; in other words, this measure compares smoothed versions of the solution and the estimate.

The results for all estimates are listed in Table 1. We observe that our Tikhonov methods typically underestimate the total emissions, which is expected when taking a zero a priori. Surprisingly, all methods from Miller et al. (2014) overestimate the total emissions, even though $\approx 5\%$ difference can be considered a good result in this setup. Estimates of the total flux from L1 POS and the Gibbs sampler are poor in this scenario.

In the local error measure, which compares grid cell by grid cell, L1 DIC POS and the transform inversion perform best. These methods come closer at addressing questions on the grid cell level but errors are still too high for accurate answers. Reasonable estimates can only be made on a coarser scale by spatially integrating grid cells. The regional measure suggests that L2 POS and the Lagrange Multiplier method also perform well on a coarser grid.

From a modeling perspective, the standard inversion is comparable to our method L2, whereas the Lagrange multiplier method and Gibbs sampler include positivity constraints and compare to L2 POS. The estimates show similar features to our estimates for L2 and L2 POS (see the supplementary information), namely rather smooth emission estimates. The spatial correlation between parameters used by Miller et al. (2014) adds to the smoothness. As already discussed for our methods, large pixel sources such as cities appear more as regional sources in the estimate. That is why these methods do not perform well on a grid cell level.

Our sparse dictionary reconstruction method and the transform inversion both estimate parameters in a different space, but the transforms are fundamentally different. For L1 DIC POS, the sparsity constraint and the dictionary with the pixel elements promote the estimation of pixel sources. For the tranform inversion, the nonlinear mapping between coefficient space and parameter space allows larger pixel emissions than the smoothing methods.

**Table 1.** Reconstruction errors measured on a local, regional and total scale (see Eqs. (16) - (18)). For the total, a negative sign means overestimation. The regional and local meausures are always positive. All measures are relative and thus without unit.

| method | rel. total | rel. regional | rel. local |
|---|---|---|---|
| L2 | 0.113 | 0.554 | 0.945 |
| L2 POS | 0.080 | 0.492 | 0.812 |
| L1 | 0.352 | 0.631 | 0.947 |
| L1 POS | 0.353 | 0.629 | 0.945 |
| L1 DIC POS | 0.051 | 0.500 | 0.747 |
| Standard Inv. | −0.052 | 0.691 | 1.039 |
| Transform Inv. | −0.057 | 0.490 | 0.683 |
| Lagrange Mult. | −0.053 | 0.523 | 0.827 |
| Gibbs Sampler | −0.273 | 0.664 | 0.957 |

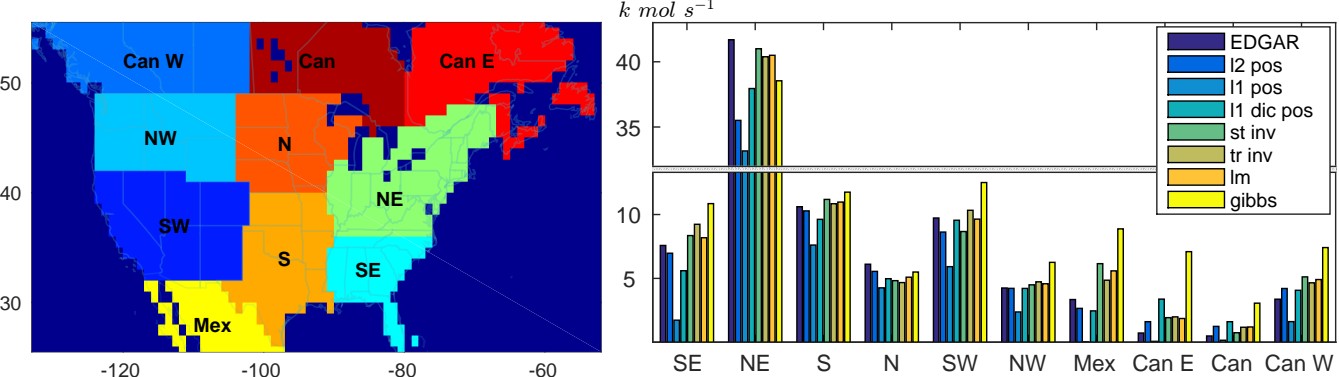

**Figure 10.** Regional EDGAR emissions and emissions estimates for the methods studied in Miller et al. (2014) and the Tikhonov reconstruction methods studied here.

### 5.3.2 Regional emission estimates

A common task is to determine the total emissions for a political or geographic region. Thus, we divided the domain into ten regions, mainly along political borders. The flux estimates for these regions are shown in Fig. 10. We see that the estimates of all methods agree for the central regions, while there are large differences in regions like Mexico and Canada. This is a consequence of data availability. Many measurement stations are located in the central regions, and the associated parameters are rather well constrained by the data. By contrast, the formulation of the a priori knowledge determines the parameters in regions with fewer observations. This results in underestimation for L2 POS and L1 POS as parameters are forced to be small respectively equal to zero. For L1 DIC POS, the background function and the broader peak shape functions included in the dictionary are used to describe the emissions in poorly-constrained regions and allow a proper estimate of the regional fluxes. Except for L1 POS and the Gibbs Sampler all methods perform comparably well for regional flux estimates.

### 5.3.3 Case study: Methane emissions from the Barnett

In a final scenario, we test the reconstruction quality of our methods for methane emissions from unconventional gas wells. We choose the Barnett shale formation in Texas because it had the highest production of any US reservoirs in summer 2008. We add a small synthetic source on top of the EDGAR fluxes and simulate noisy measurements. The synthetic emissions are inspired by the location of the formation and a recent map of well distribution (see Karion et al., 2015). The magnitude of the emissions is roughly calculated from the 2008 production rate of 85 Million $m^3$ $y^{-1}$ and a leakage of about $1.5\%$ as estimated by Zavala-Araiza et al. (2015). It is not our aim to have the most accurate emissions but to analyze the potential of the methods.

The plots in Fig. 11 show the change in the estimates induced by this additional source. Table 2 states the numbers for the spatially integrated flux change over the Barnett and the overall flux change. First, we observe that all methods underestimate the Barnett emissions. The reason for this underestimate is that all methods have low to middle sensitivity in that region (see Fig. 8), which is a consequence of data availability. However, methods differ largely in the estimated magnitude of these emissions. Best results are achieved by the transform inversion and the sparse reconstruction methods L1 POS and particularly L1 DIC POS. The other methods often adjust the total emissions adequately but have problems attributing these emissions to the Barnett. This result suggests that classical approaches lead to excessive smoothing for this application. In contrast, L1 POS is able to localize the added emissions, but it concentrates all emissions in just two grid cells within the Barnett. L1 DIC POS selects a larger and some smaller atoms from the dictionary to sparsely represent the Barnett emissions. While the source shape is not exactly reconstructed, the method nicely displays the location and the total magnitude of the emissions within the Barnett.

We should add that this scenario is not designed to favor one of these methods. The source distribution cannot be represented by a single atom in the dictionary. However, if potential source shapes like the distribution of wells were available, the sparse dictionary method would benefit from such knowledge.

**Table 2.** Results for the Barnett scenario: Estimated emissions in the Barnett region (red boxes in Fig. 11) and total flux change induced by the additional source for the methods of this study and the previous study by Miller et al. (2014).

| method | Barnett $[\text{mol s}^{-1}]$ | tot. flux $[\text{mol s}^{-1}]$ |
|---|---|---|
| L2 POS | 237.57 (36.1%) | 580.35 (88.2%) |
| L1 POS | 425.38 (64.7%) | 558.29 (84.8%) |
| L1 DIC POS | 534.82 (81.3%) | 695.66 (105.7%) |
| Standard Inv. | 323.52 (49.2%) | 625.19 (95.0%) |
| Transform Inv. | 580.59 (88.4%) | 631.76 (96.0%) |
| Lagrange Mult. | 318.63 (48.4%) | 624.35 (94.9%) |
| Gibbs Sampler | 262.26 (39.9%) | 809.23 (123.0%)* |
| true fluxes | 657.99 (100.0%) | 657.99 (100.0%) |

* The mean is approximated from a limited number of random samples from the a posteriori distribution. Thus, it slightly differs with every restart of the Gibbs sampler.

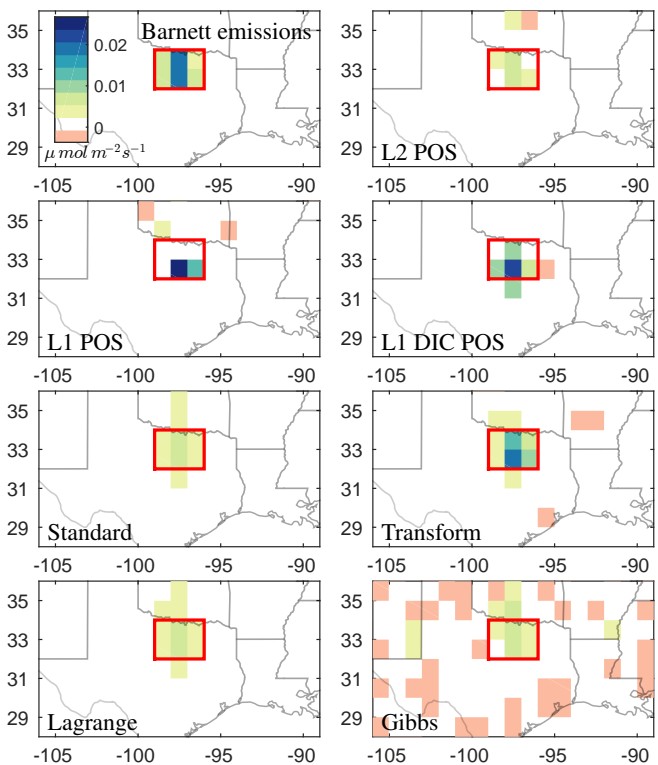

**Figure 11.** Additional methane sources in the Barnett shale gas reservoir (upper left) are added to the EDGAR emissions (see Fig. 3) and noisy data is simulated. Differences to previous reconstructions from simulated EDGAR data are shown for each method. Emissions in the red box are attributed to the Barnett.

## 6 Conclusions

This study analyzes different methods to solve inverse problems. We introduce Tikhonov regularization with the commonly used 2-norm and the sparsifying 1-norm penalty function. We show how these approaches translate to a Gaussian and a Laplacian prior, respectively, in a Bayesian inversion framework. We present a new sparse reconstruction method that enforces sparsity in a redundant dictionary representation system taylored to this application. A simple heuristic approach is applied to all methods to force nonnegative parameters. To test these methods, we consider an atmospheric inverse modeling scenario, in which we estimate methane surface fluxes for the United States from atmospheric in-situ measurements.

We find that the choice of the penalty term has a substancial influence on the estimate and is thus a crucial step when solving inverse problems. Gaussian-like priors such as the 2-norm penalty in Tikhonov regularization produce a smoothing effect. In our scenario, this characteristic means that large localized sources such as emissions from cities cannot be estimated accurately. Instead, they appear more as regional sources. In constrast, the sparse reconstruction approach can reproduce these large emitters, but it also suppresses too many small emissions to properly estimate the total flux. However, we find a simple dictionary

representation system that is able to sparsely approximate the emission field. The resulting sparse dictionary reconstruction method works equally well as established methods in determining the overall flux field and adds information on the local scale.

The Barnett case study shows the importance of such local information: While the smoothing methods recognize the additional emissions in the total flux, they cannot attribute these to the Barnett. The sparse dictionary reconstruction method and the transform inversion studied in Miller et al. (2014) perform much better in localizing these emissions. This result suggests that the standard Gaussian prior is too prohibitive towards large emitters in this application and more sophisticated models are required.

As concluded in the previous study by Miller et al. (2014), we can confirm that the positivity constraint on the flux parameters further improves the estimate. For our Tikhonov based methods, we find a heuristic approach to meet these constraints by using an iterative solver in combination with a projection step. Our iterative methods are also well-suited for large scale problems as they avoid costly numerical operations. However, an error analysis might be intractable for very large problems.

The sparsity constraint works best when the underlying signal is sparse or can be sparsely approximated. The representation of the signal in a dictionary is very flexible and can create a sparse signal for many applications. Our sparse reconstruction method is thus applicable to any inverse problem, but the dictionary would need to be adapted to suit the application. For some applications, sparsifying transforms or training data to learn a dictionary might be available. In others, finding a sparsifying dictionary might be a challenge on its own. We construct the dictionary by identifying some building blocks of the signal. The estimate can be further improved by using spatial information about sources encoded in shape functions.

In summary, the sparse reconstruction approach here is a good alternative to commonly used Gaussian priors when the emission field has many point sources or heterogeneous spatial structure. The combination of a sparsifying dictionary representation system and sparse reconstruction is a powerful tool for many inverse modeling applications.

## 7   Code and data availability

The numerical methods and the case study data are available for download. The code is written in Matlab 2014b. See the supplement for more information.

*Author contributions.*   N. H. carried out the numerical experiments, prepared and finalized the article. S. M. provided the experimental framework and was involved in the finalization of the paper. P. M., J. N., M. P. and T. W. supervised the project from a mathematical respectively an environmental physics point of view.

*Acknowledgements.*   The work was funded by the Center for Industrial Mathematics of the University of Bremen. The collaboration was supported by a research scholarship by the Deutscher Akademischer Austauschdienst (DAAD).

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
