# Peer review of "Atmospheric Inverse Modeling via Sparse Reconstruction"

_Geoscientific Model Development, 2016_

## Referee Comment (RC1) · Anonymous Referee #1 · 31 Jan 2017

Review for "Atmospheric Inverse Modeling via Sparse Reconstruction"

This manuscript presents an interesting technique for estimating surface fluxes of methane. It draws on ideas from the inverse problems literature nicely. However, there are some issues, detailed below, that need to be addressed before I can recommend the manuscript for publication.

General Comments

The style of writing is colloquial, which proved distracting for me as a reader. At several places I missed the point of the authors because of the phrasing. I suggest the authors have a senior colleague read through this draft and help them to make the writing more in line with typical scientific writing.

The idea of sparsity is fairly straightforward to grasp in terms of a basis (or dictionary), but the way it's introduced here (with the 1-norm providing the definition) is not easily connected to what is seen later. From my vantage point, it would make more sense to first point out that methane is a "point source problem", i.e. a sparsely distributed signal, and then to work forward towards inverse methods that preserve the sparsity. Additionally, it would be useful to see a simple representation of a function using a dictionary that is chosen using the two norms, to see how they perform differently. Figure 1 is too simplistic for this purpose.

In general, I'm confused by the lack of comparison to truth in your OSSEs. It's true that in real life we don't have the truth, but the metrics you've chosen leave out the true error (except for the last section comparing inverse methods). For example, you look at the sensitivity matrix, but don't actually compute the true error for the case study that is due to smoothing, which would directly show the extra smoothing for the 2-norm reconstruction.

Why is there no attempt to use the dictionary approach with the 2-norm? I would think that it would perform pretty well, particularly if the atoms were somewhat orthogonal. It's likely that I'm missing something, but this wasn't addressed in the text.

The text mixes "methods" and "results". For example, most of section 4 really belongs in the discussion of Section 3, and perhaps in a more logical style that explores the methods hierarchically. For example, introducing the 2-norm and 1-norm regularization as is done. Following this, pointing out that using a dictionary helps to focus the solution on "hot spots". Then pointing out that none of these methods enforce positive fluxes, and introducing those techniques. Then moving the material involving results all to a section 4, displaying all of the results from the different techniques side by side.

With these overarching issues, there are only a few specific comments that are pertinent to the scientific underpinnings of the paper:

Page 5, Line 22: What is the reasoning that the 1-norm targets "only a few of the

nonzero components"? Surely we could cook up a counterexample to this statement.

Page 8, Line 21: How are you selecting a basis from the dictionary? I expect that this procedure will have an impact on the final solution. I don't think subsampling a dictionary will make the solution less sparse, but it's hard to know for sure.

Page 9, Line 12: "If no uncertainty analysis is carried out" This is a pretty big blow to the technique, as emissions estimates have no value without error bounds. A quick survey of the flux inversion literature will show how variable the uncertainty estimates from different methods are.

Page 13, Figure 5: Leaving out the largest 5% seems to avoid some of the most compelling science, as "superemitters" are thought to have the biggest impact on the methane budget by a wide margin.

Figures 4, 7, 9: Errors would be more appropriate in these plots, as they would directly show "underestimate" vs. "overestimate" and put all of the sizes on the same scale. It would also show systematic behavior for methods more clearly.

Page 18: You call 5.2.1 "Smoothing Error", but no error is actually computed. Only sensitivity. However, I agree with the spirit of your intent, and think you definitely need to add the full smoothing error arising from the difference between the truth and prior/posterior.

————————————————————

---

## Referee Comment (RC2) · Anonymous Referee #2 · 21 Mar 2017

**General comments**

The paper describes a shrinkage method to estimate methane emissions. It is clearly written and has an exposition of shrinkage, as applied to atmospheric inverse problems, that is far easier to understand than others I have encountered (mostly in image processing, under the guise of compressive sensing). I am pretty sure that the paper is the first one, in atmospheric inverse problems, that uses a dictionary of basis functions and not an orthogonal basis set. It is, I believe, one of less than half-a-dozen recent papers in atmospheric inversions that uses shrinkage, which is a key advance demonstrated by this paper.

The main contributions of this paper are as follows. Anthropogenic emission fields tend not to be smoothly distributed in space; they are rough, have hotspots and are

correlated with some aspect of human economic or social activity. Estimating these fields require bases that can accommodate roughness e.g., wavelets, as was done by Ray et al, GMD, 2016 (cited in Hase's paper). Using a non-orthogonal dictionary, as Hase et al have done, injects a much higher degree of flexibility (and efficiency) of spatial representation than is possible solely with wavelets. This is because now any shape function (basis) can be used; further, the choice of shape functions can be optimized using some covariate / prior belief of methane emissions. This is the first contribution of this paper. Of course, the dictionary no longer contains orthogonal functions, and in the presence of sparse observations, using shrinkage is a must.

Shrinkage has been known in statistical literature for some time (the famous paper on LASSO appeared in JRSS-B, 1996), but imposing non-negativity on the solutions has not been a priority. That is not the case when estimating emission fields where non-negativity is a must, and Hase et al have had to invent an updating scheme that preserves non-negativity. This is a new algorithmic development in shrinkage regression as used to infer fields and is the second contribution of this paper. Note that while Ray et al, GMD, 2016 also had to impose non-negativity, Hase et al have a rather different way of doing so.

Thus the paper is quite novel and has some significant methodological contributions.

The authors have a style of writing that is direct, and at times, the paper reads a little like a text book. Though somewhat unusual, it is welcome. It makes the exposition of their ideas very clear, which should be the prime objective of the authors.

**Specific comments**

The paper introduces L1 regularization in Sec 3.3 but never provides a physical motivation for it. It is offered as a contrast and an alternative to L2 regularization, which is quite correct mathematically. However, methane emissions are expected to be "spotty" (or distributed in space in a rough manner) and a sparse representation is necessary. L1 regularization can enable such a representation by picking out appropriate atoms

from the dictionary. This is an intuition that is driven by physics and could be mentioned in the paper. Perhaps a paragraph in Sec. 3.3 somewhere or in the Conclusions would suffice. Further, in the same paragraph they could describe, in words, how L1 regularization enforce sparsity (by forcing as many elements of **x** to zero as possible [in Eq. (6)]). In contrast, L2 regularization exaggerates the large elements of **x** and concentrates on reducing them, producing smooth solutions.

In Sec. 4.6 the authors mention smoothing errors, but provide sensitivity analysis instead. Their argument is that a sensitivity analysis is the best one can do in real-life situations where the true emission field is not known. That is true. However, since this paper is a synthetic data study, these smoothing errors could be calculated. This would be akin to the errors computed in Table 2 (Barnett shale study), but without introducing noise into the synthetic observations (i.e., all the errors are due to smoothing and none due to measurement noise).

In Fig. 5, the authors plot fluxes (**x**) and show how a Laplacian distribution is a better prior than a Gaussian. However, they do not plot the top 5% of the fluxes, presumably because, being outliers, they do not fit either of the priors. Not plotting the superemitters could detract from their contention that a Laplacian prior is a better choice. Perhaps the authors could plot a Fig. 5(b) with the outliers and use that to illustrate the following facts:

- Using fluxes in each grid cell is not a good idea. They don't honor the prior, due to the existence of super-emitters.

- A different representation of the fluxes, which honors the prior, is a better idea (i.e., the fluxes are sparse in the representation used)

- Capturing the outliers could be done by augmenting the representation using the atoms (like the ones in Fig. 6). This could be a good illustration of why a dictionary of non-orthogonal functions is needed.

**Corrections, typos etc.**

1. Abstract, Pg 1, line 6: "applied math" should be "applied mathematics". Also, "developped" has 1 'p'.

2. Pg 1, line 24: "... small noise on the measurements can be "largely" amplified by the inversion ..." - largely can be omitted.

3. Pg 2, line 20: "carbondioxid" emissions .... - carbon dioxide

4. Pg 5, line 6: "...it prevents "oszillations" .... - oscillations...

5. Pg 5, line 22: ".... Which are called sparse "solutions". - "solutions" is missing

6. Pg 8, line 20: The projection step for the dictionary is "so" difficult because .... - Omit so

7. Pg 10, line 2: ....access to the noise characteristics, which is "a" common case - 'a' is missing.

8. Pg 10, line 22: "Realistic" synthetic methane emissions .... - Remove realistic? Synthetic data is never quite realistic

9. Pg 14, line 13: ... industrial facilities and mining "all" do not extend more .... - remove all?

10. Pg 15, line 25: The numerical computation requires to solve one reconstruction ....- Change to "The numerical computation requires *one* to solve *a* reconstruction ....."

11. Pg 19, line 7: The interpretation is "a" slightly different for ...- Remove 'a'

12. Pg 19, line 18: ...the method is not sensitive to data "at all" - Second use of 'at all' in the same sentence. Please omit.

13. Pg 22, line 10: "adequatly " : wrong spelling.

---

## Author Comment (AC1) · 24 May 2017

**Atmospheric Inverse Modeling**
**via Sparse Reconstruction**

**Final response of the authors**

N. Hase, S. M. Miller, P. Maaß, J. Notholt, M. Palm, and T. Warneke

First of all, we thank the topical editor for taking charge of this review process and the reviewers for their feedback and suggestions. In a first part, we comment on the general changes to the manuscript. We address all comments of the reviewers in the second part of this final response. The final part is the updated version of the manuscript with differences highlighted in red and blue colors.

**1 General changes to the manuscript**

There are a couple of general changes to the manuscript, which we briefly explain in the following.

- **Updated numerical results:**
  In the manuscript we formulate the atmospheric inverse modeling problems (Eqs. (L2), (L1) and (L1 DIC)) as optimization problems. We solve these problems using iterative methods. We found that the maximum number of iterations was insufficient for some calculations. Consequently, we increased the iteration limit, recalculated all numerical results and updated the figures and tables in the manuscript. These changes mainly appear in Sect. 4.3 and 5.

- **Total, smoothing and measurement error:**
  Both reviewers mentionend it could be helpful to include figures showing the exact total, smoothing and measurement errors. We followed this advice and included a new section (5.2.3) with graphics showing and analyzing these quantities. Sections 3.7, 4.6 and 5.2 were updated accordingly.

- **Motivation for sparsity:**
  We revised some of the phrasings to make the introduction of sparsity and sparse

dictionary reconstruction easier to the reader unfamiliar with these concepts. This update leads to changes particularly in Sect. 3.3, 4.3 and 4.4.

- **Updated histogram plot:**
  Mistakenly, we excluded the largest emissions from the calculations. The updated plot (Fig. 5) is based on full data set without removing any outliers.

**2 Author's comments to the reviews**

*Anonymous Referee #1*

*Review for "Atmospheric Inverse Modeling via Sparse Reconstruction"*

*This manuscript presents an interesting technique for estimating surface fluxes of methane. It draws on ideas from the inverse problems literature nicely. However, there are some issues, detailed below, that need to be addressed before I can recommend the manuscript for publication.*

*General Comments*

*The style of writing is colloquial, which proved distracting for me as a reader. At several places I missed the point of the authors because of the phrasing. I suggest the authors have a senior colleague read through this draft and help them to make the writing more in line with typical scientific writing.*

> Thank you for bringing this issue to our attention. We have read through the manuscript with this comment in mind and tried to revise the writing to be less colloquial whereever possible. It was our aim to present the content of this study in the most understandable form. Opinions may differ on our style of writing (see e.g. comment from Referee #2). It would be very helpful to us to point out specific paragraphs that are hard to follow.

*The idea of sparsity is fairly straightforward to grasp in terms of a basis (or dictionary), but the way it's introduced here (with the 1-norm providing the definition) is not easily connected to what is seen later.*

> Page 5, line 22 reads: 'The constraint aims at solutions of only few nonzero components, which are called sparse [solutions].' (we added 'solutions' as suggested by Referee #2)
> Vectors with few nonzero components are defined to be sparse. The sparsity constraint, i.e. the 1-norm, only helps finding such solutions, but sparsity is not defined via this constraint. In Sect. 3.3 we reformulated the paragraph

following this definition and those that introduce the concept of sparse dictionary reconstruction trying to better connect both ideas.

*From my vantage point, it would make more sense to first point out that methane is a "point source problem", i.e. a sparsely distributed signal, ...*

One could argue that methane fluxes are sparse, but we do not make such an assumption here. Instead, we assume sparsity in a dictionary representation system. A dictionary is able to provide a sparse approximation to many signals that are non-sparse in the standard grid cell representation (see e.g. [Candès et al., 2011, Starck et al., 2004, Elad, 2010]). Hence, methane fluxes do not need to be sparse for our method to work, but they could.
To be more clear about the sparse dictionary reconstruction approach, we added further explanation following Eq. 7 in Sect. 3.3. Also, we pointed out that we do not regard the inverse problem as a sparse reconstruction problem in Sect. 4.3.

*... and then to work forward towards inverse methods that preserve the sparsity. Additionally, it would be useful to see a simple representation of a function using a dictionary that is chosen using the two norms, to see how they perform differently. Figure 1 is too simplistic for this purpose.*

Unfortunately, we could not catch the idea of this comment. Figure 1 is an illustration commonly used in the inverse problem literature (see e.g. [Elad, 2010]) to demonstrate how the 1-norm penalty promotes sparsity. The effects of the different penalties on the flux field can be seen in Fig. 4 and 7. We think that after the update of the figures the difference between a smooth and a sparse flux field is easier to see in Fig. 4.
If still required, we would need further explanation on this comment.

*In general, I'm confused by the lack of comparison to truth in your OSSEs. It's true that in real life we don't have the truth, but the metrics you've chosen leave out the true error (except for the last section comparing inverse methods). For example, you look at the sensitivity matrix, but don't actually compute the true error for the case study that is due to smoothing, which would directly show the extra smoothing for the 2-norm reconstruction.*

We think that the different features of the reconstruction approaches are best seen plotting the estimates in Fig. 4 and 7. However, as this issue was raised by both reviewers we decided to add a section (5.2.3) showing and evaluating the exact total, smoothing and measurement errors.
For the interpretation of the smoothing error it is important to keep in mind that the objective is the minimization of the total error. The regularization parameter balances measurement and smoothing error. Both errors and the

balance between them may be very different for each method. A smaller smoothing error is thus no indication for a better estimate. Such details are explained in Sect. 3.7, 4.6 and 5.2.

*Why is there no attempt to use the dictionary approach with the 2-norm? I would think that it would perform pretty well, particularly if the atoms were somewhat orthogonal. It's likely that I'm missing something, but this wasn't addressed in the text.*

Large methane emitters make the EDGAR flux field rough in some locations. Such rough fields have larger 2-norm than fields that are smooth. The minimal 2-norm is thus not the ideal constraint to estimate such fields. The 1-norm constraint promotes sparse flux fields with large emitters. Sparse reconstruction selects the nonzero parameters from the full set of parameters. The dictionary approach takes advantage of this selection ability. The method can select either an atom that is spatially larger or an atom that is more concentrated or both if needed. In contrast, the 2-norm reconstruction does not have this selection property. The 2-norm is smallest if all coefficients are of the same size. Thus, a 2-norm dictionary reconstruction would tend to describe the flux field using all atoms from the dictionary. The pixel atoms are the most localized atoms already. By using spatially expanded atoms the estimate will get even smoother.

To further explore this issue, we have run experiments using the same dictionary as used with 1-norm constraint for the 2-norm constraint. The estimate is slightly worse than the estimate using the standard grid cell representation system with the 2-norm. This result is expected because the 2-norm does not select atoms. An improvement can be expected once an atom of a dictionary highly correlates with the solution. This concept is applied in the geostatistical inversion approach (see [Michalak et al., 2004]), as mentioned in Sect. 4.4. However, such atoms are constructed using additional information not used in this study.

We ran another experiment to examine the selection property of the sparsity constraint: We included an atom in the dictionary that is able to explain all emissions larger than 0.1 $\mu$ $mol$ $m^{-2}$ $s^{-1}$ perfectly and used reconstruction with 1-norm and 2-norm penalty in the dictionary representation. The estimate greatly improved for the 1-norm, but there is only very small improvement for the 2-norm. This result is due to the differing characteristics of the 1- and 2-norm. The 1-norm heading for sparse solutions selects the most apropriate atoms to describe the solution. The solution is sparser using this atom describing the large emitters than without, and the 1-norm chooses a solution that adeptly describes large emitters. In constrast, the 2-norm does not select atoms. A large coefficient in one atom results in a larger 2-norm than spreading the energy of a signal as equally as possible among all matching atoms. As a result, the 2-norm underestimates large emitters and instead smears the emissions across a broader region.

In the new version of the manuscript we pointed out the selection property of the sparsity constraint and that this property is vital for the dictionary approach. Also, we mention that the 2-norm does not have such a property.

*The text mixes "methods" and "results". For example, most of section 4 really belongs in the discussion of Section 3, and perhaps in a more logical style that explores the methods hierarchically. For example, introducing the 2-norm and 1-norm regularization as is done. Following this, pointing out that using a dictionary helps to focus the solution on "hot spots". Then pointing out that none of these methods enforce positive fluxes, and introducing those techniques. Then moving the material involving results all to a section 4, displaying all of the results from the different techniques side by side.*

One could argue that Sections 4.2 and 4.3 are misplaced. However, it was our intention to guide the reader to these new concepts of inverse modeling. We think that these preliminary results serve as a motivation for the construction of a dictionary described in Section 4.4.

*With these overarching issues, there are only a few specific comments that are pertinent to the scientific underpinnings of the paper:*

*Page 5, Line 22: What is the reasoning that the 1-norm targets "only a few of the nonzero components"? Surely we could cook up a counterexample to this statement.*

To clarify, the 1-norm targets all components. In underdetermined problems, as the simple example displayed in Fig. 1, the minimum 1-norm solution will in (almost) all cases set as many components to zero as compatible with the data. Often, this coincides with the sparsest solution.
In inverse problems we often face underdetermined problems because one needs to select a solution from a set of possible solutions that explain the data up to the noise. However, if all components of the solution are sufficiently constrained by the data, then the minimum 1-norm solution will be non-sparse.
To sum it up, the 1-norm penalty aims at sparse solutions.
To be more clear on this, we reformulated the paragraphs introducing sparse reconstruction in Sect. 3.3.

*Page 8, Line 21: How are you selecting a basis from the dictionary? I expect that this procedure will have an impact on the final solution. I don't think subsampling a dictionary will make the solution less sparse, but it's hard to know for sure.*

In our case, the dictionary contains the basis functions of the pixel basis. This submatrix of the dictionary is invertible and is used to apply the correction for negative emissions in each iteration. This procedure creates iterates that are less sparse, because coefficients that were zero prior to the nonnegativity

update may be corrected. Often, these corrections are very small and they disappear after some iterations due to shrinkage, but some will remain. Certainly, the choice of the atoms used for the correction will have an impact on the final coefficients $c^*$, but the final emission estimate $x^* = Dc^*$ is negligibly different from our experience.

We further justify our approach with the following argument. We have three different constraints on our estimate: Positivity, measurement data and sparsity. These constraints might work against one another. Using a projection strictly enforces nonnegativity. As long as Morozov's discrepancy principle is fulfilled the emissions will also always explain the data up to the noise. As a result, the most flexible constraint in our setup is the sparsity, because it is the most uncertain constraint.

The nonnegativity update for the dictionary reconstruction was described in the supplement information. We briefly answer the above question in the manuscript and refer the reader to the supplement for details in Sect. 4.5. In that section we also explained the hierachy of the constraints.

*Page 9, Line 12: "If no uncertainty analysis is carried out" This is a pretty big blow to the technique, as emissions estimates have no value without error bounds. A quick survey of the flux inversion literature will show how variable the uncertainty estimates from different methods are.*

There are a lot of inverse modeling approaches for which uncertainty bounds are difficult to estimate or are computationally prohibitive for even modestly-sized problems. These include many non-linear problems like adjoint based 4DVAR inversions (see e.g. [Wecht et al., 2014, Turner et al., 2015]) and problems with complex, non-Gaussian prior distributions. Such priors include positivity constraints (see e.g. [Miller et al., 2014, Michalak, 2008]) and sparsity exploiting formulations (see e.g. [Ray et al., 2015]). In all of these cases, the only feasible option is to come up with approximate uncertainties or a work-around borrowed from a different approach. That is what a number of the above studies have done and is what we have done here. This approach is not ideal but is common throughout the inverse modeling literature.

Some inverse problems are formulated in a deterministic way and that is where Tikhonov regularization originates. In Sect. 3.6 we have shown how Tikhonov regularization relates to the Bayesian inversion approach. In some situations such as the linear Gaussian case, it might be beneficial to make use of this relation. The conjugate gradient method we use is known to be a very efficient solver for linear inverse problems.

*Page 13, Figure 5: Leaving out the largest 5% seems to avoid some of the most compelling science, as "superemitters" are thought to have the biggest impact on the methane budget by a wide margin.*

This is a good point. At first, we excluded the 5% largest emissions because they led to a lot of empty bins in the histogram. This was misleading. We agree that we are particularly interested in these super-emitters.

We included all of the data in an updated version of Fig. 5. The new results still favor the use of a Laplacian distribution.

*Figures 4, 7, 9: Errors would be more appropriate in these plots, as they would directly show "underestimate" vs. "overestimate" and put all of the sizes on the same scale. It would also show systematic behavior for methods more clearly.*

As mentioned above, we decided to show fluxes instead of errors to show the characteristics of the estimates related to each approach. However, in the new version of the manuscript error plots are included in Sect. 5.2.3.

*Page 18: You call 5.2.1 "Smoothing Error", but no error is actually computed. Only sensitivity. However, I agree with the spirit of your intent, and think you definitely need to add the full smoothing error arising from the difference between the truth and prior/posterior.*

As mentioned we included Sect. 5.2.3 to show and analyze the total, smoothing and measurement error.

*Interactive comment on Geosci. Model Dev. Discuss., doi:10.5194/gmd-2016-256, 2016.*
* * *
*The paper describes a shrinkage method to estimate methane emissions. It is clearly written and has an exposition of shrinkage, as applied to atmospheric inverse problems, that is far easier to understand than others I have encountered (mostly in image processing, under the guise of compressive sensing). I am pretty sure that the paper is the first one, in atmospheric inverse problems, that uses a dictionary of basis functions and not an orthogonal basis set. It is, I believe, one of less than half-a-dozen recent papers in atmospheric inversions that uses shrinkage, which is a key advance demonstrated by this paper.*

*The main contributions of this paper are as follows. Anthropogenic emission fields tend not to be smoothly distributed in space; they are rough, have hotspots and are correlated with some aspect of human economic or social activity. Estimating these fields require bases that can accommodate roughness e.g., wavelets, as was done by Ray et al, GMD, 2016 (cited in Hase's paper). Using a non-orthogonal dictionary, as Hase et al have done, injects a much higher degree of flexibility (and efficiency) of spatial representation than is possible solely with wavelets. This is because now any shape function (basis) can be used; further, the choice of shape functions can be optimized using some covariate / prior belief of methane emissions. This is the first contribution of this paper. Of course, the dictionary no longer contains orthogonal functions, and in the presence of sparse observations, using shrinkage is a must.*

*Shrinkage has been known in statistical literature for some time (the famous paper on LASSO appeared in JRSS-B, 1996), but imposing non-negativity on the solutions has not been a priority. That is not the case when estimating emission fields where non-negativity is a must, and Hase et al have had to invent an updating scheme that preserves non-negativity. This is a new algorithmic development in shrinkage regression as used to infer fields and is the second contribution of this paper. Note that while Ray et al, GMD, 2016 also had to impose non-negativity, Hase et al have a rather different way of doing so.*

*Thus the paper is quite novel and has some significant methodological contributions.*

*The authors have a style of writing that is direct, and at times, the paper reads a little like a text book. Though somewhat unusual, it is welcome. It makes the exposition of their ideas very clear, which should be the prime objective of the authors.*

*Specific comments*

*The paper introduces L1 regularization in Sec 3.3 but never provides a physical motivation for it. It is offered as a contrast and an alternative to L2 regularization, which is quite correct mathematically. However, methane emissions are expected to be "spotty" (or distributed in space in a rough manner) and a sparse representation is necessary. L1 regularization can enable such a representation by picking out appropriate atoms from the dictionary. This is an intuition that is driven by physics and could be mentioned in the paper. Perhaps a paragraph in Sec. 3.3 somewhere or in the Conclusions would suffice.*

> Sect. 3 describes the methods from an applied mathematical viewpoint. Sect. 4 specifies how these mathematical concepts are used in the atmospheric inverse modeling problem considered in the manuscript. We reviewed both sections trying to better describe how sparse reconstruction works in a less mathematical way. For the sparse dictionary reconstruction a physical interpretation is hard to give before the atoms of the dictionary are specified in Sect. 4.4. We tried to be more clear by adapting the first paragraphs of Section 4.4.

*Further, in the same paragraph they could describe, in words, how L1 regularization enforce sparsity (by forcing as many elements of x to zero as possible [in Eq. (6)]). In contrast, L2 regularization exaggerates the large elements of x and concentrates on reducing them, producing smooth solutions.*

We expanded Sect. 3.3 and the subtext of Fig. 1 to be more clear on this.

*In Sec. 4.6 the authors mention smoothing errors, but provide sensitivity analysis instead. Their argument is that a sensitivity analysis is the best one can do in real-life situations where the true emission field is not known. That is true. However, since this paper is a synthetic data study, these smoothing errors could be calculated. This would be akin to the errors computed in Table 2 (Barnett shale study), but without introducing noise into the synthetic observations (i.e., all the errors are due to smoothing and none due to measurement noise).*

> As mentioned above, we have included a new section (5.2.3) showing and analyzing the total, measurement and smoothing error.
> In the Barnett shale study, basically all errors originate from smoothing because the noise is the same as in the US setup. At this point we have to mention that the sparse reconstruction methods are nonlinear. This means that the ability of reconstructing a deviation, such as the one introduced in the Barnett, might be different for each emission field and in the presence of or without noise.

*In Fig. 5, the authors plot fluxes (x) and show how a Laplacian distribution is a better prior than a Gaussian. However, they do not plot the top 5% of the fluxes, presumably because, being outliers, they do not fit either of the priors. Not plotting the super-emitters could detract from their contention that a Laplacian prior is a better choice. Perhaps the authors could plot a Fig. 5(b) with the outliers and use that to illustrate the following facts:*

- *Using fluxes in each grid cell is not a good idea. They don't honor the prior, due to the existence of super-emitters.*

- *A different representation of the fluxes, which honors the prior, is a better idea (i.e., the fluxes are sparse in the representation used)*

- *Capturing the outliers could be done by augmenting the representation using the atoms (like the ones in Fig. 6). This could be a good illustration of why a dictionary of non-orthogonal functions is needed.*

> At first, we did not use the full flux data in the histogram to avoid many empty bins. This was misleading. The large emissions are essential in the reconstruction instead of being undesired outliers. Thus, they should be respected in the

histogram analysis as well. In the updated version, we included all data.
As expected, we observe even wider shaped Gaussian and Laplacian fits. The
fits still underline that a Laplacian distribution comes closer at explaining the
EDGAR data than a Gaussian distribution.

*Corrections, typos etc.*

Thank you for detailed proofreading. We have fixed all of the typos and
corrections below.

1. *Abstract, Pg 1, line 6: "applied math" should be "applied mathematics". Also, "developped" has 1 'p'.*

2. *Pg 1, line 24: ... small noise on the measurements can be "largely" amplified by the inversion ... - largely can be omitted.*

3. *Pg 2, line 20: "carbondioxid" emissions ... - carbon dioxide*

4. *Pg 5, line 6: ... it prevents "oszillations" ... - oscillations.*

5. *Pg 5, line 22: ... Which are called sparse "solutions". - "solutions" is missing*

6. *Pg 8, line 20: The projection step for the dictionary is "so" difficult because ... - Omit so*

7. *Pg 10, line 2: ... access to the noise characteristics, which is "a" common case - 'a' is missing.*

8. *Pg 10, line 22: "Realistic" synthetic methane emissions ... - Remove realistic? Synthetic data is never quite realistic*

9. *Pg 14, line 13: ... industrial facilities and mining "all" do not extend more ... - remove all?*

10. *Pg 15, line 25: The numerical computation requires to solve one reconstruction ... - Change to "The numerical computation requires one to solve a reconstruction ..."*

11. *Pg 19, line 7: The interpretation is "a" slightly different for ... - Remove 'a'*

12. *Pg 19, line 18: ... the method is not sensitive to data "at all" - Second use of 'at all' in the same sentence. Please omit.*

13. *Pg 22, line 10: "adequatly" : wrong spelling.*

*Interactive comment on Geosci. Model Dev. Discuss., doi:10.5194/gmd-2016-256, 2016.*

[revised manuscript text omitted]

---

## Referee Report (RR1)

General Comments:  The analysis has been much improved since the last version, but it appears that some of the conclusions were hastily written, as they're not supported by the figures (see below).

Additionally, the text is still very colloquial and needs to be corrected to be more readable. There is a mixture of tenses, sometimes in the same paragraph.  For example, Page 17, Line 4-5: "For our experiments we decided not to include atoms that were constructed from EDGAR or geostatistical data. We will use a pixel basis".

Thanks to the authors for adding the error analysis in addition to the sensitivity analysis.  It's interesting that they don't quite agree (i.e. the L2 and L1 POS DIC are not all that different in errors), and this difference needs to be explained better in Section 5.2.  As it stands, it's difficult to connect the sensitivity, smoothing and actual errors in the three subsections.

Specific Comments:

Page 3, Lines 13-14: It's incorrect to say that the particles "travel backward in time".  Rather they sample the adjoint of the atmospheric transport, of which time is a dimension.  This is not the only paper to make this simplification, but it's confusing given the role of diffusion, etc in transport.  Also, the footprint is the surface influence on the measurement, rather than airmasses as it's stated here.

Page 3, Lines 20-22: This sentence at the end of the paragraph feels a bit out of place, since it's not an exhaustive list of the open problems in AIM, but rather two selected examples.  I suggest to leave out these examples and state that this method addresses only the issue of representation of solutions in the inverse problem.

Page 4, Line 4-5.  I think you mean that the dimensions of the two spaces are not the same.  Are you making a claim about the sizes of n and m?  Throughout the paper, you use the word "realistic", though that's not a precise word.  The last sentence in this paragraph should be rephrased, since each norm will give a unique solution, irregardless of how "realistic" it is.

Page 4, Line 14: The parameters are sensitive to the noise, rather than the data, in the way that you've defined the terms on Line 12.

Page 5, Line 1: "best include this information".  This seems to be a hanging fragment.

Page 5, Line 9: It may be clearer to the reader to explain what you mean by "the origin".  I'm not sure what you mean here by "oscillations".  Is it the dipole behavior in the optimized fluxes in the absence of smoothness constraints?  This needs a bit more development to be clear.

Page 5, Line 15: "and vice versa" Do you mean rougher estimates with negative correlation?  What would rougher mean?  Larger dipoles, I assume, but I haven't ever seen

negative correlations in background covariance matrices personally, and I expect that this would cause instability in the estimation problem, which assumes positive definite matrices.

Page 9, Line 29: "often straightforward to calculate analytically" - this is true only for a small set of distributions, of which the Gaussian is the prime example.  However, it's not the case that most modern flux inversion techniques estimate the posterior uncertainty using these formulae, as the covariance matrices in question are of very high rank. Typically variational or ensemble techniques construct estimates using Monte Carlo methods.

Page 10, Line 12: "a smaller smoothing error results in a greater measurement error. The smallest total error is expected when both terms are approximately balanced."  I'm not sure what you're referencing here.  Is this a general principle?  If so, please provide a reference, or give an example.'

Page 10, Line 25-26: What does it mean that assumptions are hard to guarantee?  Are typical state vectors and models not sufficiently smooth/bounded/…?

Page 21, Line 29-30: Can you mark the location of the single large point source in Figure 8?  The maps don't make this obvious at all, and look like the flux field in Figure 3, rather than a single large point source.  Maybe I'm misunderstanding?

Page 23, Line 31: Why would the L1 POS have a larger smoothing error?  Earlier text points to the success of this technique for targeting pixel sources, so this result is confusing.

Figure 9: It's not clear from this figure that L2 POS isn't the best method overall, as the dipoles seem to be smallest in this figure, even though the overall MSE is larger than the L1 DIC POS method.  Similarly, the smoothing error isn't obviously better in the bottom panel than the top panel.

Page 26, Line 5-7: This conclusion is much too strong.  For the regional fluxes, the standard inversion is as accurate as the L1 POS DIC inversion, particularly in the large emitting regions.  It's difficult to pick a clear winner between all of the different methods in this case.

---

## Author Response (AR2)

**Atmospheric Inverse Modeling via Sparse Reconstruction**

**2nd final response of the authors**

N. Hase, S. M. Miller, P. Maaß, J. Notholt, M. Palm, and T. Warneke

First of all, we thank the topical editor again for taking charge of this review process and the reviewer for his/her feedback and suggestions. In a first part, we comment on the general changes to the manuscript. We address the comments of the reviewer in the second part. The final part is the updated version of the manuscript with differences highlighted in red and blue colors.

**1 General changes to the manuscript**

Only minor changes were made to the manuscript. Most changes correct for faulty English language.

**2 Author's comments to the reviews**

*Anonymous Referee #1*
*Submitted: 19 July 2017*
*Review for "Atmospheric Inverse Modeling via Sparse Reconstruction"*

*General comments:*

*The analyis has been much improved since the last version, but it appears some of the conclusions were hastily written, as they're not supported by the figures (see below)*

> We comment on this in the specific comments part.

*Additionally, the text is still very colloquial and needs to be corrected to be more readable. There is a mixture of tenses, sometimes in the same paragraph. For example, Page 17, Line 4-5: "For our experiments we decided not to include atoms that were constructed from EDGAR or geostatistical data. We will use a pixel basis."*

> The text has been proofread and corrected by a native speaker, which led to minor corrections. As mentioned in the previous final response, it is 'our aim to present the content of this study in the most understandable form. Opinions may differ on our style of writing.' (see 'Final response of the authors', P. 2)

*Thanks to the authors for adding the error analysis in addition to the sensitivity analysis. It's interesting that they don't quite agree (i.e. the L2 and L1 POS DIC are not all that different in errors), and this difference needs to be explained better in Section 5.2. As it stands, it's difficult to connect the sensitivity, smoothing and actual errors in the three subsections.*

> We comment on this in the specific comments part.

*Specific comments:*

*Page 3, Lines 13-14: It's incorrect to say that the particles "travel backward in time". Rather they sample the adjoint of the atmospheric transport, of which time is a dimension. This is not the only paper to make this simplification, but it's confusing given the role of diffusion, etc in transport. Also, the footprint is the surface influence on the measurement, rather than airmasses as it's stated here. ...*

> The authors of the STILT model use the formulation 'backward in time' to describe their modeling approach (see [Lin et al., 2003]). We adopted this formulation. Obviously, this description is a modeling concept rather than a description of the true processes. We want to give the reader an intuition of how the model works, particularly readers who may not be familiar with model adjoints.

*... Also, the footprint is the surface influence on the measurement, rather than airmasses as it's stated here.*

> Page 3, Lines 12-14 read: "STILT will release an ensemble of imaginary particles at the time and location of an atmospheric measurement. The particles then travel backward in time and indicate where air masses were located before reaching the measurement location. STILT then uses the distribution of these particles to compute an upwind [surface] influence on the measurement, called footprint. The footprint quantitatively relates the surface fluxes to the atmospheric measurement [...]."

We corrected the tense, added the units for the footprint and replaced "influence" by "surface influence" to be more precise. We are not sure what is meant by the reviewer's comment.

*Page 3, Lines 20-22: This sentence at the end of the paragraph feels a bit out of place, since it's not an exhaustive list of the open problems in AIM, but rather two selected examples. I suggest to leave out these examples and state that this method addresses only the issue of representation of solutions in the inverse problem.*

We use the atmospheric inverse modeling approach

$$z = y + b = Ax + b,$$

where $z$ are the atmospheric in-situ measurements consisting of background concentrations $b$ and enhancements thereof $y$. The transport model $A$ describes the relationship between surface fluxes $x$ and enhancements $y$. In our case, $A$ is determined by the STILT model footprints.
The interpretation of atmospheric measurements $z$ based on this modeling approach thus involves $A$, $b$ or $x$. Our study focusses only on one aspect of AIM, i.e. the determination of surface fluxes $x$, given $A$ and $y$ $(= z - b)$. With the chosen formulation we intend to express that the determination of $A$ and $b$ are open problems, too. Based on this equation the list of problems is complete, though not detailed.
However, we added that the list of problems may not be complete.

*Page 4, Line 4-5. I think you mean that the dimensions of the two spaces are not the same. Are you making a claim about the sizes of n and m? ...*

Measurement space $Y$ and parameter space $X$ can be any suitable space. In most applications one will choose $\mathbb{R}^m$ and $\mathbb{R}^n$, respectively. A theoretical physicist might use specific functional spaces for $X$ and $Y$, which might require a deeper mathematical theory to solve the inverse problem. We specify the dimensions for the test problem in Sect. 4.2.

*... Throughout the paper, you use the word "realistic", though that's not a precise word. The last sentence in this paragraph should be rephrased, since each norm will give a unique solution, irregardless of how "realistic" it is.*

By 'realistic' we mean that the value for a quantity could be true. It meets our expectations without having determined it. The phrase 'realistic/unrealistic' appears 4 times in the article. We find that it properly describes what we intend to express in each instance. We are open to specific suggestions for reformulations.

*Page 4, Line 14: The parameters are sensitive to the noise, rather than the data, in the way that you've defined the terms on Line 12.*

The inverse mapping maps from data space $Y$ to parameter space $X$. If the problem is ill-posed, the output, i.e. the parameters, is sensitive to the input, which is any form of data. This can be noiseless data, noisy data or noise. We avoid a formulation that uses 'sensitive to noise' because noise is not an isolated input.

*Page 5, Line 1: "best include this information". This seems to be a hanging fragment.*

We corrected this aspect.

*Page 5, Line 9: It may be clearer to the reader to explain what you mean by "the origin". I'm not sure what you mean here by "oscillations". Is it the dipole behavior in the optimized fluxes in the absence of smoothness constraints? This needs a bit more development to be clear.*

The 'origin' refers to the origin or zero point of a coordinate system. We have clarified this point in the text.
Unstable inversions produce estimates of oscillating behaviour. These oscillations originate from the fact that the coefficients to high frequent singular vectors are perturbed most by the noise in ill-posed problems. This article does not present the formulation of a solution using singular vectors. We added a reference that explains this fact (see [Hansen, 2010]) and reformulated the phrase to be more specific.

*Page 5, Line 15: "and vice versa" Do you mean rougher estimates with negative correlation? What would rougher mean? Larger dipoles, I assume, but I haven't ever seen negative correlations in background covariance matrices personally, and I expect that this would cause instability in the estimation problem, which assumes positive definite matrices.*

A covariance matrix can include negative correlations and be positive definite, e.g.
$$Q = \begin{pmatrix} 1 & -\frac{1}{2} \\ -\frac{1}{2} & 1 \end{pmatrix}.$$
The corresponding weighting matrix $L_a$ can then be calculated by $Q^{-1} = \alpha(L_a^t L_a)$, i.e.
$$Q^{-1} = \frac{4}{3} \begin{pmatrix} 1 & \frac{1}{2} \\ \frac{1}{2} & 1 \end{pmatrix}.$$

Assuming $\alpha = \frac{4}{3}$, it remains

$$L_a = \begin{pmatrix} 1 & \frac{1}{2} \\ 0 & \frac{\sqrt{3}}{2} \end{pmatrix} \quad \text{or} \quad L_a \approx \begin{pmatrix} 0.9659 & 0.2588 \\ 0.2588 & 0.9659 \end{pmatrix} \quad \text{or} \quad L_a = \begin{pmatrix} 1 & \frac{1}{2} \end{pmatrix},$$

where the first matrix is calculated using the Cholesky decomposition and the second matrix is the principle matrix square root. The third matrix is a rectangular matrix decomposition that avoids redundant information.

We agree that positive correlations are used more frequently than negative correlations. However, this argument does not rule out the use of negative correlations in cases where such information is available.

The underconstrained test problem

$$x^* = \arg\min_{x \in \mathbb{R}^2} \|\begin{pmatrix} 1 & 0 \end{pmatrix} x - \begin{pmatrix} 1 \end{pmatrix}\|_2^2 + \alpha\|L_a x\|_2^2$$

illustrates the consequences of correlating weighting matrices. For $\alpha = 1$ we have

$$\begin{aligned} \text{positive correlation} \quad & L_a = \begin{pmatrix} 1 & -1 \\ -1 & 1 \end{pmatrix} & x^* = \begin{pmatrix} 1 \\ 1 \end{pmatrix}, \\ \text{no correlation} \quad & L_a = \begin{pmatrix} 1 & 0 \\ 0 & 1 \end{pmatrix} & x^* = \begin{pmatrix} \frac{1}{2} \\ 0 \end{pmatrix}, \\ \text{negative correlation} \quad & L_a = \begin{pmatrix} 1 & 1 \\ 1 & 1 \end{pmatrix} & x^* = \begin{pmatrix} 1 \\ -1 \end{pmatrix}. \end{aligned}$$

The test problem is set up such that only the first component of the solution is constrained by the measurement. The second component is only constrained by the penalty term. In the uncorrelated case, there is a trade-off between data and penalty reducing the first component. For the correlated cases, the matrices create a null space. Still, the minimizer is unique in both cases. Positive correlation increases the smoothness compared to the uncorrelated case. Negative correlation reduces the smoothness.

*Page 9, Line 29: "often straightforward to calculate analytically" - this is true only for a small set of distributions, of which the Gaussian is the prime example. However, it's not the case that most modern flux inversion techniques estimate the posterior uncertainty using these formulae, as the covariance matrices in question are of very high rank. Typically variational or ensemble techniques construct estimates using Monte Carlo methods.*

This statement was wrong without further specifications and confusing given the explanations on Page 9, Lines 23 - 26. We corrected both paragraphs accordingly.

*Page 10, Line 12: "a smaller smoothing error results in a greater measurement error. The smallest total error is expected when both terms are approximately balanced." I'm not sure what you're referencing here. Is this a general principle? If so, please provide a reference, or give an example.'*

This is a general principle in inverse problem theory (see e.g. [Hansen, 2010], Ch. 5). We added the reference to the manuscript.

*Page 10, Line 25-26: What does it mean that assumptions are hard to guarantee? Are typical state vectors and models not sufficiently smooth/bounded/...?*

Several studies have studied error bounds $E$ in the form

$$\|x^+ - x^*\| \le E,$$

where $x^+$ is the true solution and $x^* = R(y^\delta)$ is the estimate of a reconstruction method from noisy data $y_\delta = Ax^+ + \delta$. Such bounds can only be installed if the noise $\delta$ and the true solution $x^+$ meet certain assumptions. A common assumption for the noise is

$$\|\delta\| \le \bar{\delta},$$

where $\bar{\delta}$ is called the noise level. Assumptions on the true solution vary in the studies. A classical approach assumes an abstract smoothness, i.e. $x^+ \in \mathcal{N}(A)^\perp$ or smoother (see e.g. [Tautenhahn, 1998]). It can be hard to show that a physical quantity meets these assumptions.

Error bounds have also been established for sparse reconstruction methods (see e.g. [Jin and Maass, 2012],[Elad, 2010]). In finite dimensional spaces, sparse reconstruction methods require that the solution $x^+$ is $k$-sparse. This means that at most $k$ elements of the solution vector are nonzero. How large $k$ is depends on properties of the forward model $A$ (see e.g. [Elad, 2010]). Typically, it is hard to guarantee that $x^+$ is $k$-sparse. However, it might have a sparse approximation.

*Page 21, Line 29-30: Can you mark the location of the single large point source in Figure 8? The maps don't make this obvious at all, and look like the flux field in Figure 3, rather than a single large point source. Maybe I'm misunderstanding?*

As explained in the caption and in the text above (see Page 21, Line 19) Fig. 8 shows the diagonal of the sensitivity matrix. Each column of this matrix shows the response of the inverse model to data that includes an additional pixel source. Ideally, the response should be exactly this pixel source. The diagonal of the matrix shows how much of the pixel source would be seen in that exact pixel. Methods that produce smoother emission fields are expected to have smaller values on the diagonal, because pixel sources are smoothed.

We have reformulated Lines 29-30 on Page 21 because the formulation appeared to be misleading.

*Page 23, Line 31: Why would the L1 POS have a larger smoothing error? Earlier text points to the success of this technique for targeting pixel sources, so this result is confusing.*

As mentioned on Page 23, Lines 17-20 "it is misleading to look at either the smoothing or measurement error without the other. For ill-posed inverse problems, a small smoothing error comes at the expense of a larger measurement error and vice versa. A well chosen regularization parameter balances both errors such that the total error is minimized.
We illustrate this behaviour using classical Tikhonov regularization, i.e.

$$x^* = \arg \min_{x \in \mathbb{R}^n} \frac{1}{2}\|Ax - y^\delta\|_2^2 + \frac{\alpha}{2}\|x\|_2^2$$
$$= \left(A^t A + \alpha I\right)^{-1} A^t y^\delta$$

with the notations from the manuscript. For $y^\delta = Ax^+ + \delta$ we have the total error

$$x^* - x^+ = \left(A^t A + \alpha I\right)^{-1} A^t y^\delta - x^+$$
$$= \left(A^t A + \alpha I\right)^{-1} A^t (Ax^+ + y^\delta) - x^+$$
$$= \underbrace{\left(A^t A + \alpha I\right)^{-1} A^t Ax^+ - x^+}_{\text{smoothing error}} + \underbrace{\left(A^t A + \alpha I\right)^{-1} A^t \delta}_{\text{measurement error}}.$$

This method is known to produce smooth solutions. However, for the choice $\alpha = 0$ the smoothing error is zero. Thus, the method is similar to unconstrained least squares, which largely amplifies the noise, leading to a large measurement error. On the other hand, we could choose a very large value for $\alpha$. This will produce the estimate $x^* \approx 0$. As a result the measurement error is zero but the smoothing error is equal to $x^+$. The regularization parameter is ideally chosen such that the total error is minimized. Whether smoothing or measurement error dominate for this optimal regularization parameter depends on the problem and the method used.

With this in mind, one can observe for L1 POS that the smoothing error is reduced in some pixels with larger sources (e.g. Salt Lake City). However, this improvement in some locations does not balance the errors made in other locations and causes a larger smoothing error.

*Figure 9: It's not clear from this figure that L2 POS isn't the best method overall, as the dipoles seem to be smallest in this figure, even though the overall MSE is larger than the L1 DIC POS method. Similarly, the smoothing error isn't obviously better in the bottom panel than the top panel.*

We agree that it is hard to see visually which method performs best. That is the reason for calculating the mean squared error. We use the mean squared error because it is a standard metric in image analysis.

*Page 26, Line 5-7: This conclusion is much too strong. For the regional fluxes, the standard inversion is as accurate as the L1 POS DIC inversion, particularly in the large emitting regions. It's difficult to pick a clear winner between all of the different methods in this case.*

We agree that the conclusions were too strong as written. With the exception of L1 POS and the Gibbs Sampler all methods have a comparable average performance in the estimation of regional fluxes. We changed the conclusion in Sect. 5.3.2 accordingly.

**3 Updated manuscript**

The updated manuscript is on the next pages. Changes in the text compared to the previous version are marked in colors.

[revised manuscript text omitted]